# ATTRIBUTED GRAPH CLUSTERING VIA GENERALIZED QUATERNION REPRESENTATION LEARNING

## ABSTRACT

Clustering complex data in the form of attributed graphs has attracted increasing attention, where appropriate graph representation is a critical prerequisite for accurate cluster analysis. However, the Graph Convolutional Network will homogenize the representation of graph nodes due to the well-known over-smoothing effect. This limits the network architecture to a shallow one, losing the ability to capture the critical global distribution information for clustering. Therefore, we propose a generalized graph auto-encoder network, which introduces quaternion operations to the encoders to achieve efficient structured feature representation learning without incurring deeper network and larger-scale parameters. The generalization of our method lies in the following two aspects: 1) connecting the quaternion operation naturally suitable for four feature components with graph data of arbitrary attribute dimensions, and 2) introducing a generalized graph clustering objective as a loss term to obtain clustering-friendly representations without requiring a pre-specified number of clusters $k$. It turns out that the representations of nodes learned by the proposed Graph Clustering based on Generalized Quaternion representation learning (GCGQ) are more discriminative, containing global distribution information, and are more general, suiting downstream clustering under different $k$s. Extensive experiments including significance tests, ablation studies, and qualitative results, illustrate the superiority of GCGQ. The source code is temporarily opened at `https://anonymous.4open.science/r/ICLR-25-No7181-codes`.

## 1 INTRODUCTION

Learning clustered distributions of data in an unsupervised way is a fundamental data analysis process in artificial intelligence tasks. As graph data contain richer relational information among data objects (also called nodes interchangeably), clustering complex data represented in the form of graphs has attracted increasing attention, where graph representation Shi & Malik (2000); Ng et al. (2001); Von Luxburg (2007) is critical to clustering accuracy. Some recent works Pan et al. (2018); Liu (2022) further consider attribute values of graph nodes that reflect their inherent similarity relationship to achieve a more information-comprehensive clustering.

To perform attributed graph clustering, conventional representation learning approaches Ren et al. (2020; Feb. 2021) usually adopt multiple kernel functions for node embedding. However, this type of approach involves the non-trivial selection of kernels and is vulnerable to the curse of dimensionality. Under such circumstances, end-to-end deep graph representation learning based on Graph Convolution Network (GCN) Kipf & Welling (2017) is considered an effective way to enhance the performance of attributed graph clustering. GCN and its variants Bowman et al. (2015); Wang et al. (2019); Zhang et al. (2022); Mrabah et al. (2022) can simultaneously learn the embedding of graph structure and attribute values to achieve a more informative representation.

To explore the cluster distribution of data from a global perspective, relationships among nodes that span far in the graph are also critical. Although stacking more graph convolutional layers may theoretically help extract long-span information, the high overlap of over-hopped nodes will tend to homogenize the node representations, which is known as the widely discussed *over-smoothing* effect Li et al. (2018). Existing solutions for mitigating such effect can be categorized into training tricks Zhou et al. (2021); Rong et al. (2020); Zhao & Akoglu (2020), dynamic hopping strategies Rusch

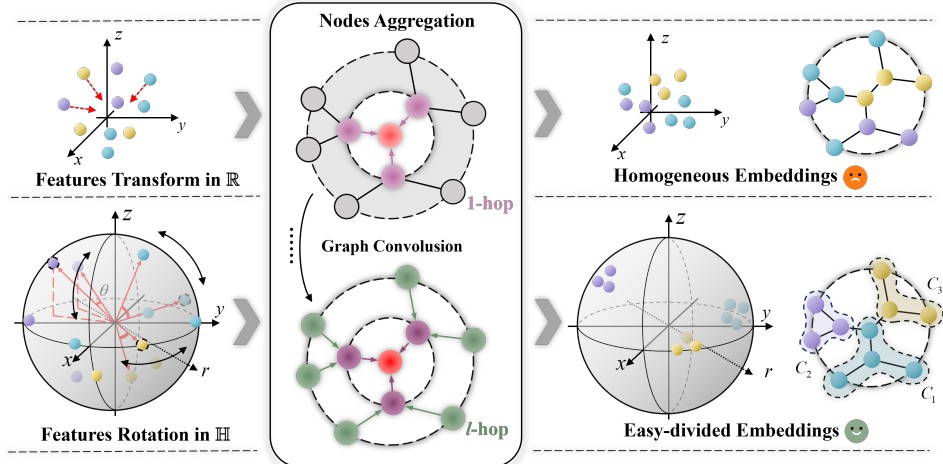

Figure 1: Vanilla graph encoders (upper) vs. quaternion graph encoders (lower). After the node information aggregation through several hops, nodes represented in real-value space $\mathbb{R}$ by vanilla graph encoders tend to be homogeneous due to the "over-smoothing" and "over-dominating" effects. By contrast, the four views of data are flexibly rotated in a hyper-complex space $\mathbb{H}$ by the quaternion graph encoders to facilitate representation learning with a higher degree of learning freedom.

et al. (2022); Eliasof et al. (2021), residual connections Chen et al. (2020); Xu et al. (2018), and more powerful representation enhancement paradigms, e.g., contrastive learning Yang et al. (2023a;b).

However, most existing over-smoothing solutions originate from processing attribute-free graphs, which naturally overemphasize the topological information of graphs and tend to overlook the attribute information. That is, the embeddings of topology-adjacent but attribute-dissimilar nodes will be similar due to the information aggregation dominated by the graph topology. Such an *over-dominating* effect will somewhat lead to the loss of discriminative attribute information that helps distinguish nodes from different clusters. For nodes with inconsistent topological relationships and attribute similarities, this effect will further degrade the clustering performance. Therefore, how to simultaneously cope with the over-smoothing and over-dominating effects to obtain discriminative node representations is the key to attributed graph clustering.

This paper proposes a novel and concise Quaternion Representation Learning (QRL) model for attributed graph data by introducing the quaternion operation to the encoders. The whole model inherits the framework of graph auto-encoder Kipf & Welling (2016), but projects any dimensional input attributes into four views corresponding to one real and three imaginary parts of the quaternion. Then the quaternion graph encoders efficiently perform structural transformation to the attribute views and aggregate the graph topological information. Figure 1 intuitively compares the principles of linear encoders and quaternion encoders under the scenario of attributed graph clustering. In the quaternion encoding process, since each attribute view is transformed as a whole, the node description information they contain can be retained to a greater extent to relieve the over-dominating effect. From a more macro perspective, the "wide" four-view projection and the corresponding encoding layers ensure the representation capability with a "shallow" network, which naturally circumvents the over-smoothing problem brought by "deep".

To adapt the learned representation to different clustering tasks, we integrate the graph reconstruction loss with the graph clustering objective for training. Since clustering does not involve benchmark node labels, users often understand a dataset by performing clustering using different $k$s to observe the clusters in different granularities. To be compatible with such an actual usage scenario, a generalized clustering objective is integrated into the loss so that the model can be trained without specifying $k$, and general representations suitable for different clustering granularities can be obtained. It turns out that the proposed Graph Clustering based on Generalized QRL (GCGQ) can well aggregate the attribute and graph information to produce discriminative and general embeddings. Moreover, thanks to the efficient Hamilton product Zhang (1997) of quaternions, GCGQ does not incur extra computation costs compared to the advanced graph clustering methods. Theoretical

analysis and extensive experiments on various real benchmark graph datasets have illustrated the efficiency, efficacy, and superiority of GCGQ. The main contributions are summarized in three-fold:

- We propose a generalized representation learning framework for attributed graph clustering. It bridges the gap between graphs with arbitrary attribute dimensions and the four-part quaternion algebra. It also connects the representation learning to clustering tasks through the design of a general graph clustering objective loss. To the best of our knowledge, this is the first attempt to: 1) introduce QRL to unsupervised learning tasks, and 2) realize clustering-friendly representation learning without requiring a specified $k$.

- To simultaneously address the over-smoothing and over-dominating effects, we propose to perform multi-view projection and quaternion graph convolution encoding. Such design allows for a shallower and wider network to circumvent the over-smoothing effect without sacrificing the representation ability, and preserves the attribute information.

- The proposed GCGQ is efficient and resolves the ill-posed assumption of current deep clustering, i.e., the "true" number of clusters $k$ is known in advance for model training. That is, GCGQ can provide universal representations without repeatedly training the model at different cluster granularities controlled by $k$. Such characteristic is crucial for practical clustering applications and data distribution understanding.

## 2 RELATED WORK

### 2.1 DEEP ATTRIBUTED GRAPH CLUSTERING

Deep attributed graph clustering that partitions connected nodes described by attribute values into compact clusters has attracted much attention in recent years. Benefiting from the powerful representation reconstruction ability of Auto-Encoder (AE) Vincent et al. (2008) and Variational Auto-Encoder (VAE) Bowman et al. (2015), GAE and VGAE Kipf & Welling (2016) are proposed with graph convolution operator for graph reconstruction. Inspired by the success of GAE, recent works further improve it by introducing the attention mechanism Wang et al. (2019) and the adversarial learning mechanism Pan et al. (2018). To perform more accurate graph clustering, some recent works like EGAE and R-GAE Zhang et al. (2022); Mrabah et al. (2022) propose to customize the representations to be clustering-friendly by optimizing both reconstruction and clustering objectives during the model training. Most recently, contrastive learning Yang et al. (2023a), as a powerful learning capability enhancement paradigm, has also been introduced to graph clustering. It adopts clustering as a proxy task for data augmentation, and generates more discriminative node embeddings. Later, a learnable reversible perturb-recover proxy task is further considered in contrastive graph clustering Yang et al. (2023b). It more reliably preserves semantic information in the augmentation, and thus achieves more satisfactory clustering performance.

### 2.2 QUATERNION REPRESENTATION LEARNING

Quaternion is a four-dimensional extension of complex numbers with a completed algebra foundation. Since the Hamilton product Zhang (1997) efficiently facilitates the interaction between the four parts of quaternions through the quaternion vector rotation upon the three imaginary axes, the quaternion operator is considered promising to enhance the representation learning ability Parcollet et al. (2020), especially for the data with natural relations among its feature tuples, e.g., the three channels of colored image Zhu et al. (Sep. 2018); Zheng et al. (2023); Parcollet et al. (May 2019) and the 3D sound signals Comminiello et al. (2019). The work Zhang et al. (2019) is considered the first attempt to introduce the powerful QRL for knowledge graph embedding. Almost all the existing usage of quaternion is in supervised scenarios. Moreover, the input data are usually with inherent tuple or multiple feature components corresponding to the three imaginary parts of the quaternion, and the feature components are also interdependent, especially suitable for representation learning using quaternion rotation operations. However, how to inherit the merits of quaternion to unsupervised learning tasks has been relatively unexplored.

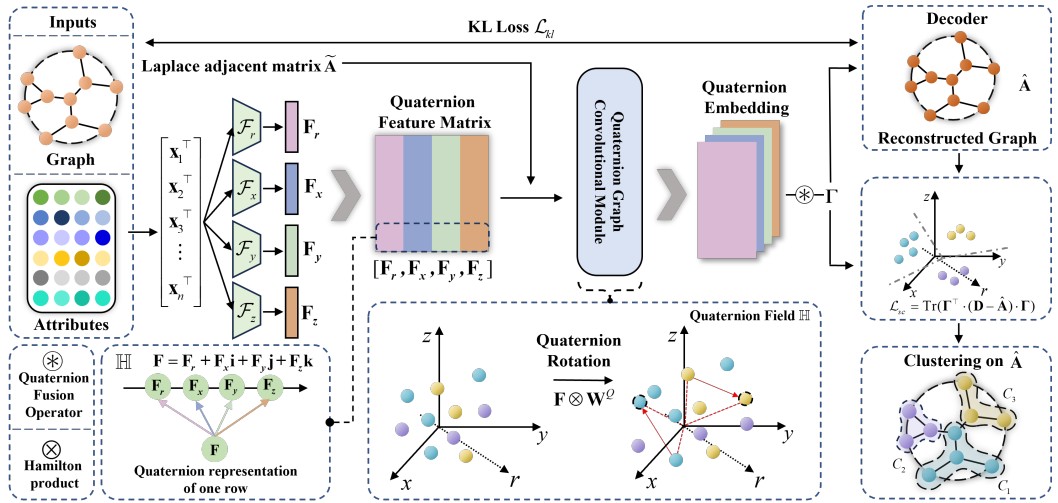

Figure 2: Overview of GCGQ. Given attributed graph $G = \{\mathbf{A}, \mathbf{X}\}$, the attributes $\mathbf{X}$ are first projected into four views to form a feature quaternion $\mathbf{F} = \mathbf{F}_r + \mathbf{F}_x\mathbf{i} + \mathbf{F}_y\mathbf{j} + \mathbf{F}_z\mathbf{k}$. Then $\mathbf{F}$ is encoded with the local graph structure by a quaternion graph convolutional module. The graph clustering-friendly embedding $\mathbf{\Gamma}$ learned according to the joint graph reconstruction Kullback-Leibler (KL) loss $\mathcal{L}_{kl}$ and the graph clustering loss $\mathcal{L}_{sc}$ is finally obtained, which is utilized for clustering.

## 3 PROPOSED METHOD

We first provide the preliminaries, and then introduce the proposed graph clustering method GCGQ. The overview of GCGQ is demonstrated in Figure 2.

### 3.1 PRELIMINARIES

A quaternion is denoted as $Q = r + x\mathbf{i} + y\mathbf{j} + z\mathbf{k}$ where $r$ is the real part and $x, y, z$ are the imaginary parts. The Hamilton product between two quaternions $Q_1 = r_1 + x_1\mathbf{i} + y_1\mathbf{j} + z_1\mathbf{k}$ and $Q_2 = r_2 + x_2\mathbf{i} + y_2\mathbf{j} + z_2\mathbf{k}$ can be denoted as $Q_1 \otimes Q_2$, which follows the laws of association and distribution, but does not follow the law of commutativity. Benefiting from the orthogonality among imaginary axes, the essence of the product is to rotate $Q_1$ according to $Q_2$ in the hyper-complex space $\mathbb{H}$ spanned by the three imaginary axes. This is considered a critical characteristic expected by representation learning, especially for the learning of naturally coupled feature components.

Given an undirected attributed graph $G = \{\mathbf{A}, \mathbf{X}\}$ with node attributes $\mathbf{X} \in \mathbb{R}^{n \times d}$ and adjacency matrix $\mathbf{A} \in \mathbb{R}^{n \times n}$, where $n$ and $d$ are the number of nodes and dimensions, respectively. The attribute matrix can also be denoted into the form of $n$ nodes $\mathbf{X} = [\mathbf{x}_1, \mathbf{x}_2, \cdots, \mathbf{x}_n]^\top$, which will be grouped into $k$ clusters by partitioning the graph $\mathbf{A}$ into $k$ non-overlapping sub-graphs $\{G_1, G_2, ..., G_k\}$. A degree matrix $\mathbf{D} \in \mathbb{R}^{n \times n}$ is a diagonal matrix reflecting the connectivity of each node, which is formed by $\mathbf{D}_{ii} = \sum_{j=1}^{n} \mathbf{A}_{ij}$. The symmetric normalized Laplacian matrix of $\mathbf{A}$ that actually participates in the representation learning is denoted as $\tilde{\mathbf{A}} = \mathbf{D}^{-\frac{1}{2}}\mathbf{L}\mathbf{D}^{-\frac{1}{2}}$, where $\mathbf{L} = \mathbf{I} + \mathbf{A}$ is a self-loop adjacency matrix and $\mathbf{I}$ is a unit matrix. The self-loop and normalization operations are to prevent nodes from ignoring their own information and the nodes with higher degrees from dominating the information passing during the graph convolution.

### 3.2 GENERALIZED QUATERNION REPRESENTATION LEARNING

**FVP: Four-View Projection** Unlike most existing QRL scenarios that the datasets are with tuple feature components (e.g., RGB images), attributed graph data are with different numbers of attributes and various graph structures. To leverage QRL in attributed graph representation learning, we design a learnable projection mechanism to project the attributes $\mathbf{X}$ into four views. Such a mechanism acts to lift the tuple restriction of input features in QRL and also leverages the Hamilton product for efficient coupling learning of the attributes.

Specifically, four independent initial MLPs are utilized to project the nodes represented by the attributes $\mathbf{X} = [\mathbf{x}_1, \mathbf{x}_2, \cdots, \mathbf{x}_n]^{\top}$ into four views $\mathbf{F}_r$, $\mathbf{F}_x$, $\mathbf{F}_y$, and $\mathbf{F}_z$, which can be written as

$$\mathbf{F}_{\triangleright} = \mathcal{F}_{\triangleright}(\mathbf{X}) = \mathbf{W}_{\triangleright}^L \mathbf{X} + \mathbf{B}_{\triangleright}^L, \ \ \triangleright \in \{r, x, y, z\}, \tag{1}$$

where $\mathcal{F}_{\triangleright}(\cdot)$ indicates an MLP opterator, $\mathbf{W}_{\triangleright}^L$ and $\mathbf{B}_{\triangleright}^L$ are the learnable parameters of an MLP. The generated four views form the feature quaternion $\mathbf{F} \in \mathbb{H}^{n \times (4 \times \hat{d})}$ as

$$\mathbf{F} = \mathbf{F}_r + \mathbf{F}_x \mathbf{i} + \mathbf{F}_y \mathbf{j} + \mathbf{F}_z \mathbf{k}, \tag{2}$$

where $\hat{d}$ is the dimensionality of the features encoded by $\mathcal{F}_{\triangleright}(\cdot)$, and each row of $\mathbf{F}$ is actually the quaternion representation of the corresponding node. By introducing a learnable weight quaternion $\mathbf{W}^Q \in \mathbb{H}^{n \times (4 \times \hat{d})}$ with the same size as $\mathbf{F}$, $\mathbf{F}$ can be projected based on $\mathbf{W}^Q$ by

$$\mathbf{F} \otimes \mathbf{W}^Q = \begin{bmatrix} \mathbf{F}_r \\ \mathbf{F}_x \\ \mathbf{F}_y \\ \mathbf{F}_z \end{bmatrix}^{\top} \begin{bmatrix} \mathbf{W}_r^Q & \mathbf{W}_x^Q & \mathbf{W}_y^Q & \mathbf{W}_z^Q \\ -\mathbf{W}_x^Q & \mathbf{W}_r^Q & -\mathbf{W}_z^Q & \mathbf{W}_y^Q \\ -\mathbf{W}_y^Q & \mathbf{W}_z^Q & \mathbf{W}_r^Q & -\mathbf{W}_x^Q \\ -\mathbf{W}_z^Q & -\mathbf{W}_y^Q & \mathbf{W}_x^Q & \mathbf{W}_r^Q \end{bmatrix}, \tag{3}$$

where $\otimes$ indicates the Hamilton product. The above $\mathcal{F}_{\triangleright}(\cdot)$ and $\otimes$ processes can convert an arbitrary-dimensional attribute set into the quaternion field, and associate different parts of the feature quaternion through their shared weights in the weight quaternion $\mathbf{W}^Q$. By tuning the weights in $\mathbf{W}^Q$, features in $\mathbf{F}$ can be efficiently transformed with capturing their couplings.

**Remark 1** *Degree of Freedom (DoF). According to Eq. (3), learnable parameters of $\mathbf{W}^Q$, i.e., $\{\mathbf{W}_r^Q, \mathbf{W}_x^Q, \mathbf{W}_y^Q, \mathbf{W}_z^Q\}$, yield 16 pairs of feature transformation to determine arbitrary rotation of $\mathbf{F}$ in hyper-complex space $\mathbb{H}$, while in real-value space $\mathbb{R}$, realizing the same transformation requires four times of parameters. Therefore, the DoF of quaternion feature transformation is four times higher than the transformation in real-value space (detailed proof is provided in Appendix D.2).*

The low parameter scale and high DoF of quaternion transformation allow a representation model to tolerate a wider structure with four-view projection MLPs to preserve more original attribute information. As a result, attributes can be amplified to offset the dominant effects caused by the subsequent graph convolution processes, and more discriminative similarity information of nodes can be embedded to boosting clustering performance.

**QGE: Quaternion Graph Encoders**   To further make a fusion of the feature quaternion $\mathbf{F}$ with the graph topology $\tilde{\mathbf{A}}$, $\mathbf{F}$ is feed forward to a quaternion graph convolutional module composed of stacked encoders, where the operation of the $l$-th encoder can be written as

$$H_l = \varphi_l(\tilde{\mathbf{A}} \cdot H_{l-1} \otimes \mathbf{W}_l^Q) \tag{4}$$

where the operation priority of $\otimes$ is higher than the matrix product. Here, $\varphi_l(\cdot)$ is the activation function and $\tilde{\mathbf{A}} \in \mathbb{R}^{n \times n}$ is the Laplacian adjacency matrix. Each encoder aggregates the $l$-hop quaternion representation of nodes according to the graph topology $\tilde{\mathbf{A}}$, to yield a more abstract-level representation $H_l$. The output embeddings of the quaternion graph convolutional module with $m$ encoders are integrated into a single matrix $\boldsymbol{\Gamma}$ by

$$\boldsymbol{\Gamma} = \mathrm{Re}(H_m) \circledast \mathrm{Im}(H_m), \tag{5}$$

where $\mathrm{Re}(\cdot)$ and $\mathrm{Im}(\cdot)$ indicate the fetch of real part and imaginary parts of $\mathbf{F}$, respectively, and the operation $\circledast$ is a quaternion fusion operator that takes an average of the four feature quaternion parts. Finally, we reconstruct the graph as $\hat{\mathbf{A}} \in \mathbb{R}^{n \times n}$ based on the embeddings $\boldsymbol{\Gamma}$ by

$$\hat{\mathbf{A}} = \boldsymbol{\Gamma} \cdot \boldsymbol{\Gamma}^{\top}. \tag{6}$$

The graph reconstruction acts as a decoder to ensure the preservation of the graph topology.

### 3.3 Clustering-Oriented Loss and Optimization

From a macro perspective of the model, the FVP and QGE modules collaboratively emphasize the attribute information in $\mathbf{\Gamma}$, and the graph reconstruction acts to adapt $\mathbf{\Gamma}$ to the graph structure to seek balanced attribute and graph consensus. To also make the reconstructed graph sparse to be graph clustering friendly, the joint loss function is designed as a combination of the graph reconstruction term $\mathcal{L}_{kl}$, spectral clustering term $\mathcal{L}_{sc}$, and regularization term $\mathcal{L}_{reg}$, which can be written as

$$\mathcal{L} = \mathcal{L}_{kl} + \alpha\mathcal{L}_{reg} + \beta\mathcal{L}_{sc}, \tag{7}$$

where $\alpha$ and $\beta$ are trade-off hyper-parameters. We adopt $\mathcal{L}_{kl}$ to quantify the reconstruction loss by

$$\mathcal{L}_{kl} = \frac{1}{n^2} \sum_{i=1}^{n} \sum_{j=1}^{n} \tilde{\mathbf{A}}_{ij} \log \frac{1}{\hat{\mathbf{A}}_{ij}}, \tag{8}$$

where $n$ is the number of nodes, $\tilde{\mathbf{A}}$ and $\hat{\mathbf{A}}$ are the original Laplacian adjacency matrix and the adjacency matrix reconstructed by Eq. (6), respectively. By minimizing $\mathcal{L}_{kl}$, consensus embeddings $\mathbf{\Gamma}$ can be achieved on the graph topology reflected by $\tilde{\mathbf{A}}$ and the learned embeddings of the node attributes indicated by $\hat{\mathbf{A}}$. The regularization term $\mathcal{L}_{reg}$ is to avoid the over-fitting of the model by restricting its complexity. The spectral clustering loss term is defined as

$$\mathcal{L}_{sc} = \mathrm{Tr}(\mathbf{\Gamma}^{\top} \mathbf{L} \mathbf{\Gamma}), \tag{9}$$

where $\mathbf{L} = \mathbf{D} - \hat{\mathbf{A}}$ is the Laplacian matrix formed based on the degree matrix $\mathbf{D}$ of the original graph structure $\mathbf{A}$ and the learned attributed graph representation $\hat{\mathbf{A}}$. Referring to the spectral clustering objective in Eq. (10), $\mathbf{\Gamma}$ of $\mathcal{L}_{sc}$ can be viewed as a relaxed node-cluster affiliation indicator matrix. The minimization of $\mathcal{L}_{sc}$ prefers a $\mathbf{\Gamma}$ that can reconstruct graph with sparser adjacency (i.e., smaller values in $\mathbf{D}$) and higher feature similarity of connected nodes (i.e., larger values in $\hat{\mathbf{A}}$), both are consistent with the spectral clustering objective.

When completing the model training, we obtain the clustering-friendly graph representation $\hat{\mathbf{A}}$ by Eq. (6), and the corresponding semi-definite Laplacian matrix $\mathbf{L} = \mathbf{D} - \hat{\mathbf{A}}$ is prepared to well-support the optimization of spectral clustering objective:

$$\arg\min_{\mathbf{H}} \mathrm{Tr}(\mathbf{H}^{\top} \mathbf{L} \mathbf{H}) \quad s.t. \ \mathbf{H}^{\top} \mathbf{H} = \mathbf{I}. \tag{10}$$

Here $\mathbf{H} \in \mathbb{R}^{n \times k}$ is the indicator matrix indicating the node-cluster affiliations, and $\mathbf{I}$ is the unit matrix. Spectral clustering solves the above problem by first computing $\mathbf{E}$, which is the $k$-smallest eigenvectors of $\mathbf{L}$. Then K-Means clustering is performed on the $n \times k$ matrix $\mathbf{E}$ by treating each of its rows as the representation of the corresponding node. Note that the user only need to give the target cluster number $k$ at this time. Please refer to Von Luxburg (2007) and Ikotun et al. (2023) for more eigenvalue decomposition and K-Means clustering details.

The whole GCGQ algorithm is summarized in Appendix B.1, and its complexity analysis is provided in Appendix B.2. It is noteworthy that $\mathbf{L}$ obtained based on $\hat{\mathbf{A}}$ is the key factor to influence the accuracy of clustering. To learn more powerful $\hat{\mathbf{A}}$, the model is designed with a higher Degree of Freedom (DoF) in feature encoding facilitated based on the quaternion product. On such basis, the training process comprehensively takes into account the attribute information by the FVP and QGE, preserves the graph topology by the graph reconstruction decoder, and customizes the general clustering-friendly representation by introducing the clustering-oriented loss $\mathcal{L}_{sc}$. It turns out that the obtained node representations $\mathbf{E}$ for K-Means clustering are with strong cluster discriminability. That is, nodes with close topological relationships and similar attribute values in the input attributed graph $G$ will have shorter Euclidean distances in $\mathbf{E}$, promising to boost clustering accuracy.

## 4 Experiment

### 4.1 Experimental Settings

**Datasets** Experiments are conducted on ten real benchmark attributed graph datasets, including CORA Sen et al. (2008), CITESEER Sen et al. (2008), DBLP Bo et al. (Apr. 2020), ACM Bo et al.

(Apr. 2020), WIKI Yang et al. (2015), FILM Liu (2022), and the four, i.e., CORNELL, WISC, UAT, and AMAP, from Liu et al. (2022). The CORA and DBLP datasets are the citation network. ACM and DBLP datasets are paper citation relationships. WIKI and FILM datasets are the relationships of Wikipedia links and films, respectively. CORNELL, WISC, UAT, and AMAP datasets are American university website links. Detailed dataset statistics are sorted in the Appendix in Table A.1.

**Training Process** All the experiments are implemented in PyTorch 1.8.0 on NVIDIA A5000 GPU, 64GB RAM. We first warm up the model by a 10-epoch training using only the KL loss $\mathcal{L}_{kl}$ and regularization loss $\mathcal{L}_{reg}$. We follow the most recent graph clustering works Yang et al. (2023a); Zhang et al. (2022); Wang et al. (2019); Tu et al. (2021); Yang et al. (2023b) to obtain the clustering performance: Each result is the average performance with standard deviation on ten implementations of the compared methods. For each implementation, the model is trained by 50 epochs. In each epoch, we train the model in four iterations and then perform clustering. The best clustering performance of the 50 epochs is chosen to be the performance of the current implementation.

**Counterparts Setup** 11 clustering methods are compared, including two traditional methods, i.e., K-Means Hamerly & Elkan (2003) and Spectral Clustering (Spectral-C, to distinguish from the internal evaluation metric SC) Shi & Malik (2000), two conventional representation learning-based clustering methods, i.e., GAE Kipf & Welling (2016) and VGAE Kipf & Welling (2016), seven state-of-the-art deep clustering methods including ARGAE and ARVGAE Pan et al. (2018), CON-VERT Yang et al. (2023b), CCGC Yang et al. (2023a), DFCN Tu et al. (2021), DAEGC Wang et al. (2019), and EGAE Zhang et al. (2022). We let K-Means directly perform clustering on the data attributes. All the other methods obtain node representations first and then implement K-means on the representations. Settings of all the compared methods and GCGQ are reported in Appendix A.2.

**Validity Metrics** Six evaluation metrics are utilized. Three external metrics Zhou et al. (2022b): Clustering Accuracy (ACC), Normalized Mutual Information (NMI), and Average Rand Index (ARI), which evaluate performance according to the data labels, are in the intervals $[0, 1]$, $[0, 1]$, and $[-1, 1]$, respectively. Three internal metrics: Silhouette Coefficient (SC) Rousseeuw (1987), Davies-Bouldin Index (DBI) Davies & Bouldin (1979), and Calinski-Harabasz Index (CHI) Caliński & Harabasz (1974), that do not rely on the labels, are in the intervals $[-1, 1]$, $[0, +\infty)$, and $[0, +\infty)$. All these metrics are commonly used by most of the compared state-of-the-art methods, and except for DBI, a higher value indicates a better clustering performance.

## 4.2 QUANTITATIVE RESULTS

We conduct four groups of quantitative experiments: 1) Compare clustering performance using external metrics to illustrate the clustering accuracy superiority of GCGQ; 2) Compare clustering performance using internal metrics under different $k$s to verify the separability and universality of the embeddings learned by GCGQ; 3) Compare execution time to validate the efficiency of GCGQ; 4) Compare different ablated versions of GCGQ to prove the effectiveness of its core modules.

**Clustering Performance Evaluated by External Metrics** Table 1 reports the clustering performance of all the compared methods by using $k$ provided by the data labels. The significance test described in Appendix A.4 is also conducted, and the results shown in Table A.2 demonstrate that GCGQ passes all the Wilcoxon signed rank tests with a confidence interval of 95% (except for the CONVERT method in terms of ACC), which validates its superiority. From Table 1, it can be observed that the proposed GCGQ outperforms the compared methods in most cases. Out of the 294 comparisons, GCGQ won 290 times, which generally demonstrates its superiority. Note that six 'N/A' cases happened when implementing ARGAE and ARVGAE on the AMAP dataset, as they suffered from gradient explosions. In the following, another three key observations are provided:

1) There are four performance groups of the compared methods: Based on the average ranks in the "AR" row, there are four groups of methods with prominent AR gaps. The K-Means and Spectral Clustering (Spectral-C) with ARs around nine belong to the first group, as they are traditional methods without representation learning. The second groups would be the GAE, VGAE, ARGAE, ARVGAE, DFCN, and DAEGC with ARs within $[5.7, 7.3]$. They are all based on the GAE and the latter two (i.e., DFCN and DAEGC) further incorporate clustering objectives for training. The third

Table 1: Clustering performance compared with existing methods. The best and second-best results on each dataset are marked in **boldface** and underline, respectively. 'N/A' indicates 'not available' due to gradient explosion. "AR" in the last row reports the average performance rank of each method.

| Dataset | Metric | K-Means | Spectral-C | GAE | VGAE | ARGAE | ARVGAE | CONVERT | CCGC | DFCN | DAEGC | EGAE | GCGQ (Ours) |
|---|---|---|---|---|---|---|---|---|---|---|---|---|---|
| ACM | ACC | 36.78±0.01 | 74.21±0.00 | 44.22±4.11 | 59.88±1.57 | 78.56±5.10 | 86.94±1.37 | 80.53±2.91 | 89.26±0.60 | 86.04±2.18 | 74.61±10.00 | 85.54±3.62 | **90.37±0.41** |
|  | NMI | 00.82±0.01 | 52.45±0.01 | 14.67±4.53 | 18.78±1.13 | 44.88±7.13 | 58.20±3.29 | 47.45±4.35 | 65.36±1.21 | 59.66±4.51 | 47.92±10.35 | 56.09±8.26 | **67.50±1.17** |
|  | ARI | 00.24±0.01 | 47.65±0.00 | 03.66±2.39 | 15.59±1.86 | 46.36±11.01 | 64.94±3.29 | 51.30±6.03 | 71.06±1.37 | 63.94±4.79 | 48.70±12.59 | 62.10±8.21 | **73.57±1.06** |
| WIKI | ACC | 25.81±0.89 | 17.46±0.35 | 33.11±2.08 | 31.73±0.75 | 28.11±1.47 | 44.47±3.66 | 51.41±1.15 | 51.29±0.84 | 43.10±3.67 | 25.38±3.35 | 47.49±1.13 | **52.95±0.88** |
|  | NMI | 22.69±1.21 | 08.84±0.16 | 31.62±1.51 | 27.25±0.38 | 23.15±1.94 | 44.13±2.65 | 48.46±0.62 | 46.19±1.01 | 38.33±2.91 | 15.15±2.63 | 43.33±1.99 | **49.26±1.32** |
|  | ARI | 02.54±0.32 | -00.30±0.09 | 05.61±0.89 | 15.63±0.79 | 06.23±1.13 | 24.44±3.24 | 28.39±1.33 | 25.50±2.72 | 17.17±3.75 | 07.68±2.25 | 28.99±1.58 | **33.77±0.87** |
| CITESEER | ACC | 26.10±1.33 | 19.56±0.01 | 32.93±3.01 | 55.10±2.19 | 44.64±7.66 | 54.37±2.96 | 62.14±1.53 | 66.31±2.27 | 42.37±2.05 | 42.66±4.74 | 58.71±3.68 | **66.57±1.14** |
|  | NMI | 06.92±1.36 | 00.31±0.00 | 20.11±2.63 | 27.92±0.86 | 19.07±6.89 | 27.54±2.85 | 34.68±1.78 | **40.45±2.68** | 23.90±1.83 | 18.79±3.56 | 33.15±2.99 | 40.36±1.21 |
|  | ARI | 00.31±1.93 | 00.08±0.00 | 04.64±2.01 | 26.78±1.68 | 16.07±7.15 | 25.11±3.66 | 34.69±1.88 | 39.12±3.36 | 19.19±2.43 | 16.81±4.38 | 31.46±4.61 | **41.43±1.94** |
| DBLP | ACC | 32.74±0.06 | 29.92±1.01 | 46.10±1.43 | 47.07±2.43 | 55.31±4.93 | 54.97±6.88 | 54.52±2.37 | 54.78±1.97 | 38.91±0.04 | 43.36±4.72 | 53.64±1.46 | **72.46±2.24** |
|  | NMI | 02.98±0.01 | 00.28±0.22 | 19.71±1.83 | 17.72±2.11 | 20.63±3.63 | 22.61±5.44 | 22.33±1.93 | 23.81±2.53 | 08.11±0.04 | 11.41±3.55 | 18.19±1.07 | **39.12±2.27** |
|  | ARI | 15.31±1.87 | 00.20±2.80 | 05.78±0.87 | 14.39±1.95 | 18.14±4.36 | 17.70±5.12 | 17.81±1.17 | 18.64±1.28 | 06.63±0.02 | 10.40±3.70 | 15.07±2.02 | **41.24±2.55** |
| FILM | ACC | 24.21±0.01 | 24.05±0.04 | 25.64±0.02 | 21.40±0.79 | 23.84±0.47 | 24.31±1.32 | **27.43±0.23** | 26.36±0.11 | 25.91±1.64 | 24.61±0.33 | 22.79±0.25 | 26.81±0.65 |
|  | NMI | 00.01±0.00 | 00.11±0.01 | 00.09±0.01 | 00.07±0.01 | 00.16±0.05 | 00.22±0.39 | 00.79±0.07 | 00.15±0.01 | 00.28±0.03 | 00.09±0.03 | 00.21±0.08 | **01.47±0.22** |
|  | ARI | 00.00±0.01 | -00.14±0.02 | 00.13±0.01 | 00.01±0.02 | 00.11±0.03 | 00.31±0.51 | 01.34±0.17 | 00.24±0.05 | 00.27±0.04 | 00.15±0.10 | 00.17±0.08 | **01.78±0.26** |
| CORNELL | ACC | 42.40±0.65 | 37.81±2.34 | 38.03±1.09 | 26.66±1.16 | 36.99±2.54 | 36.55±2.65 | 41.86±2.98 | 39.61±2.09 | 39.72±1.90 | 36.28±2.11 | 39.23±0.53 | **38.25±1.84** |
|  | NMI | 02.71±0.17 | 03.69±0.62 | 05.35±0.36 | 03.25±0.97 | 06.01±1.22 | 03.19±0.56 | **09.80±2.68** | 04.89±1.04 | 03.25±0.37 | 06.83±1.36 | 06.49±0.73 | 08.86±2.51 |
|  | ARI | -02.14±0.10 | -00.15±0.55 | 02.11±0.48 | -00.06±0.54 | 02.43±1.96 | 00.85±1.18 | **06.19±3.25** | 02.07±1.06 | -01.10±1.23 | 02.15±2.08 | 03.17±0.91 | 05.48±1.32 |
| CORA | ACC | 31.14±3.76 | 24.47±0.03 | 49.47±5.76 | 63.47±0.69 | 65.96±4.12 | 66.72±3.04 | 66.34±1.80 | 72.00±1.77 | 45.94±5.80 | 45.30±5.92 | 72.11±1.35 | **75.82±1.51** |
|  | NMI | 06.67±5.28 | 01.48±0.01 | 40.86±4.81 | 45.45±0.59 | 44.75±3.69 | 48.96±2.62 | 46.84±1.68 | 55.02±1.91 | 36.46±3.44 | 25.88±4.35 | 52.89±1.17 | **59.02±1.20** |
|  | ARI | 07.83±1.69 | -00.08±0.01 | 22.49±7.27 | 39.01±0.85 | 39.52±4.55 | 42.80±2.69 | 40.13±1.67 | 49.17±2.40 | 23.95±5.52 | 20.09±5.99 | 48.49±2.16 | **55.64±3.06** |
| WISC | ACC | 42.03±2.04 | 30.31±0.11 | 42.11±1.73 | 25.77±1.34 | 36.01±2.18 | 37.17±2.58 | **47.61±1.91** | 44.14±0.86 | 40.95±5.44 | 25.38±3.35 | 37.01±2.01 | 44.46±1.58 |
|  | NMI | 06.25±1.13 | 03.73±0.01 | 08.09±0.63 | 02.60±1.40 | 11.02±2.61 | 05.35±3.26 | 09.70±3.35 | 08.39±0.45 | 07.01±0.58 | 15.15±2.63 | 11.02±1.16 | **16.31±1.66** |
|  | ARI | -03.02±1.68 | 00.02±0.01 | 02.85±0.54 | 00.03±0.48 | 05.56±1.86 | 02.02±1.63 | 04.76±2.69 | 03.60±0.90 | 04.45±0.99 | 07.68±2.25 | 06.00±1.12 | **09.92±1.38** |
| UAT | ACC | 32.69±0.12 | 32.52±0.01 | 44.55±0.07 | 37.45±3.46 | 49.36±1.30 | 41.85±1.63 | **55.18±1.34** | 47.88±2.69 | 39.33±4.72 | 52.49±1.25 | 53.10±0.79 | 53.84±0.27 |
|  | NMI | 20.63±0.63 | 03.43±0.00 | 18.61±4.94 | 17.68±0.95 | 23.33±1.71 | 15.86±2.38 | **27.31±1.18** | 20.63±2.64 | 13.95±1.67 | 21.42±1.29 | 21.80±0.85 | 24.15±1.16 |
|  | ARI | 06.42±0.44 | 01.57±0.00 | 11.61±5.90 | 14.35±0.84 | 16.76±0.68 | 10.33±2.82 | 19.46±1.90 | 12.95±1.80 | 07.28±3.16 | 21.07±1.11 | 20.77±0.64 | **22.54±0.91** |
| AMAP | ACC | 22.66±0.31 | 17.24±0.01 | 60.47±0.87 | 68.61±0.51 | N/A | N/A | 66.28±1.86 | **77.07±0.38** | 58.51±3.96 | 47.45±3.36 | 76.37±1.32 | 76.02±0.94 |
|  | NMI | 02.37±0.09 | 00.53±0.00 | 58.01±0.56 | 55.04±0.46 | N/A | N/A | 52.57±0.79 | **67.06±0.72** | 55.95±1.13 | 38.83±4.24 | 65.44±1.61 | 66.47±1.50 |
|  | ARI | 00.37±0.03 | 00.00±0.01 | 33.31±1.02 | 46.55±0.67 | N/A | N/A | 42.89±1.49 | 57.55±0.44 | 41.76±1.58 | 25.03±4.68 | 57.51±1.74 | **57.73±1.50** |
| - | AR | 9.9 | 10.7 | 7.8 | 8.2 | 7.0 | 6.4 | 3.5 | 3.6 | 7.0 | 7.6 | 4.4 | 1.5 |

group consists of CONVERT, CCGC, and EGAE, all with an AR of around 4. The proposed GCGQ surely belongs to the fourth group with an AR close to one.

2) GCGQ vs. CCGC/CONVERT: The proposed GCGQ achieves great performance improvements compared to the best-performing counterparts, which is usually the CCGC and CONVERT. Specifically, GCGQ outperforms the best-performing counterparts by 16.5%, 121.2%, 13.2%, and 29.2% on WIKI, DBLP, CORA, and WISC datasets, respectively, in terms of ARI. On most other datasets, our GCGQ also achieves considerable improvements of around 5% in comparison with the rivals. Compared to our GCGQ, the CCGC and CONVERT methods adopt a contrastive learning paradigm by treating K-Means as the proxy task. Their data augmentation effectiveness relies on the selection of proper cluster number $k$, which is a non-trivial task, because the original $k$ provided by the dataset labels is not necessarily the 'true' $k$. Accordingly, the performance of CCGC and CONVERT is relatively unstable on different datasets.

3) GCGQ vs. EGAE: EGAE adopts a relaxed K-Means to optimize the representation, which also requires a proper cluster number $k$, and thus achieves satisfactory clustering performance. Even though the training process of our GCGQ is not guided by $k$, it still stably performs the best in most cases. The reason would be that even the 'true' $k$ provided by the original dataset may still be unsuitable for the fusion of inconsistent attributes and graph topology. The use of a given $k$ can be viewed as introducing a strong hypothesis that may implicitly restrict the fitting ability of representation learning. By contrast, GCGQ adopts a relaxed clustering objective without restricting the node representations to be concentrated on $k$ potential clusters. As a result, GCGQ fosters a high DoF learning, and thus universal clustering-friendly representations can be obtained.

**Separability and Universality Evaluation of Representations** For clustering, the separability of learned representations is often evaluated by internal metrics. To also verify the effectiveness of the generalized loss of GCGQ, we compare the clustering performance under different $k$s of GCGQ with EGAE, CCGC, and CONVERT, which are the state-of-the-art counterparts that performed better in Table 1. The comparison results are shown in Figure 3, and it can be seen that the proposed GCGQ always outperforms the state-of-the-art counterparts under different $k$s w.r.t. all the metrics. Such results simultaneously prove the outstanding separability and universality of the embeddings learned by the GCGQ in general. More specific observations are as follows: 1) As the value of $k$ increases, the performance of GCGQ gradually degrades. This is reasonable because the internal metrics mainly measure the separation between clusters and the compactness within clusters. When there are many clusters (large $k$), the ability of the metrics to discriminate the capabilities of different representations will naturally weaken. A more extreme case is that when $k = n$, all the compared

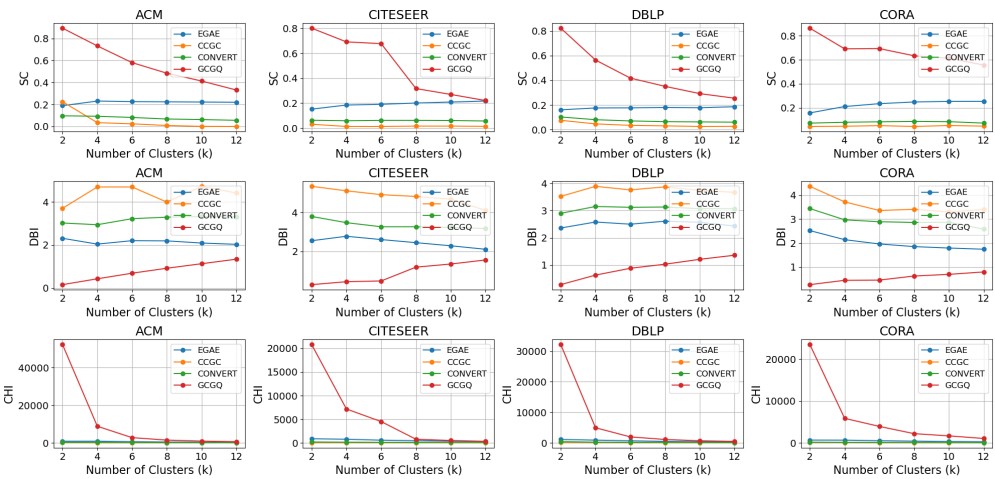

Figure 3: Clustering performance comparison using internal metrics under different $k$s. For the SC and CHI metrics, the higher the better. For the DBI metric, the lower the better.

methods will have similar performance; 2) The $k$s in Figure 3 all cover the $k$s provided by the dataset labels in Table A.1. At the $k$s provided by the labels, GCGQ performs better than the other methods, proving that its representation has better separability for more accurate clustering.

**Efficiency Evaluation** Corresponding to the four datasets and three advanced methods in the previous experiment as shown in Figure 3, we also compare their execution time with the proposed GCGQ averaged on the six different $k$ values. The execution time comparison is visualized in Figure 4. It can be observed that the execution time of the compared methods is higher than that of GCGQ. This is because GCGQ can learn general representations to support the clustering under different $k$s without retraining the model, and thus its model training time averaged on the six runs of the clustering is relatively lower. By contrast, the other three methods need to train the model according to the specified $k$, which causes more training overhead.

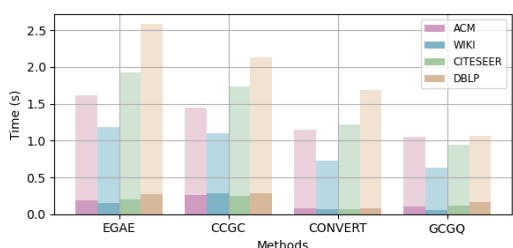

Figure 4: Average execution time on different $k$ values. Different colors indicate the average execution time on different datasets. Deep and shallow colors indicate the execution time of model training and clustering.

**Ablation Studies** Table 2 compares the clustering performance of GCGQ with: 1) Baseline: A model composed of one MLP and two stacked GCN encoders, 2) GCGQ w/o FVP: GCGQ with a frozen FVP module, 3) GCGQ w/o QGE: GCGQ with its QGE module replaced by the conventional GCN encoders. By comparing GCGQ with GCGQ w/o FVP, GCGQ w/o QGE, and Baseline, the effectiveness of the designed FVP, the necessity of introducing

Table 2: Ablation study of the key modules of GCGQ. The best and second-best results in terms of each validity metric are marked in **boldface** and underline, respectively.

| Dataset | Baseline | | | GCGQ w/o FVP | | | GCGQ w/o QGE | | | **GCGQ** | | |
|---|---|---|---|---|---|---|---|---|---|---|---|---|
| | ACC | NMI | ARI | ACC | NMI | ARI | ACC | NMI | ARI | ACC | NMI | ARI |
| ACM | 89.34 | 64.54 | 70.92 | 89.90 | 65.80 | 72.27 | 84.53 | 56.38 | 60.85 | **90.37** | **67.50** | **73.57** |
| WIKI | 51.60 | 49.15 | 32.43 | 51.57 | 47.91 | 31.99 | 51.64 | 48.66 | 32.45 | **52.95** | **49.26** | **33.77** |
| CITESEER | 65.73 | 40.24 | 40.72 | 66.21 | **40.44** | 41.28 | 66.57 | 40.38 | 41.35 | **66.57** | 40.36 | **41.43** |
| DBLP | 67.89 | 35.20 | 35.48 | 71.41 | 38.15 | 39.73 | 67.26 | 36.59 | 35.71 | **72.46** | **39.12** | **41.24** |
| FILM | 26.95 | 1.12 | 1.76 | 27.41 | 1.28 | 1.97 | **27.70** | **1.51** | **2.01** | 26.81 | 1.47 | 1.78 |
| CORNELL | 35.85 | 6.87 | 3.69 | 36.99 | 6.52 | 4.51 | 36.61 | 6.52 | 3.93 | **38.25** | **8.86** | **5.48** |
| CORA | 72.73 | 55.84 | 51.44 | 73.12 | 55.13 | 50.61 | 72.11 | 55.35 | 49.68 | **75.82** | **59.02** | **55.64** |
| WISC | 40.92 | 13.54 | 8.03 | 43.03 | 16.09 | 9.37 | **44.74** | **17.21** | **10.45** | 44.46 | 16.31 | 9.92 |
| UAT | 53.68 | 23.69 | 21.87 | 53.71 | 23.77 | 22.36 | **54.37** | 23.26 | 22.19 | 53.84 | **24.15** | **22.54** |
| AMAP | 73.23 | 61.13 | 53.81 | 74.69 | 63.65 | 55.53 | 74.98 | 64.34 | 56.37 | **76.02** | **66.47** | **57.73** |

QGE, and the adaptability of FVP and QGE, can be validated, respectively. Three observations are provided below: 1) GCGQ performs better than the Baseline in 29 out of 30 comparisons, clearly

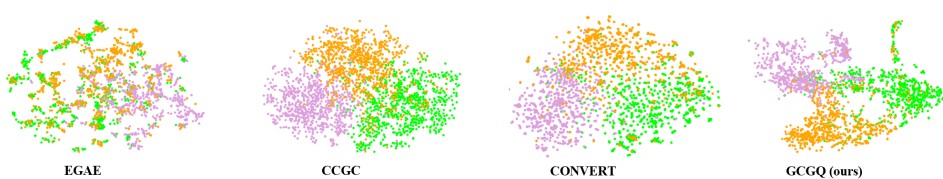

Figure 5: $t$-SNE visualization of ACM dataset represented by different methods.

illustrating the adaptability of FVP and QGE. 2) GCGQ performs better than GCGQ w/o FVP in 27 out of 30 comparisons. This evidently indicates that FVP is a necessary pre-phase of QGE. GCGQ w/o FVP makes the four MLPs unlearnable, and thus FVP degrades to a random projection of the input attributes, which surely loses the ability to provide more suitable feature quaternions for QGE. 3) GCGQ outperforms GCGQ w/o QGE in 22 out of 30 comparisons. This generally indicates that QGE is effective in aggregating the node information and preventing the over-dominating effect. Without the Hamilton product in QGE, GCGQ w/o QGE cannot facilitate a high DoF learning of attribute information, and thus the graph topology may dominate the node information aggregation. That is, the embeddings of two very dissimilar connected nodes may be homogeneous in the final embeddings, which may severely hamper the clustering accuracy.

### 4.3 QUALITATIVE RESULTS

To intuitively show the representation effectiveness of GCGQ, we visualize the distributions of embeddings generated by the state-of-the-art EGAE, CCGC, CONVERT, and our GCGQ on the ACM dataset in Figure 5. The 2-D plots are generated using $t$-SNE Van der Maaten & Hinton (2008) and we use different colors to mark the label-provided clusters. Intuitively, CCGC, CONVERT, and our GCGQ perform better with more separable clusters against the EGAE. The reason would be that CCGC and CONVERT adopt contrastive augmentation, and GCGQ adopts quaternion rotation, to effectively enhance the learning capability of their representation models. Since GCGQ performs structural rotation of the four views of attributes, the global distribution of nodes is better preserved, and thus the embedding clusters of GCGQ are even more separable compared to that of CCGC and CONVERT. The GAE-based EGAE method is probably over-dominated by the graph topology as it does not specifically emphasize the preservation of attribute information.

### 5 CONCLUDING REMARKS

This paper proposes a novel attributed graph clustering method called GCGQ. It leverages the advantages of the efficient Hamilton product of quaternions to simultaneously tackle the over-smoothing and over-dominating issues that bottleneck the clustering performance. Through generalized design, a representation learning model composed of learnable FVP and QGE is formed for clustering-friendly representation learning. The FVP module bridges the gap between any dimensional attributes and the four-part quaternion operation of QGE, and these two modules collaboratively enhance: 1) the learning capability of the model, and 2) the preservation of attribute information. The generalized clustering objective loss guides the model to learn universal representations with high DoF without restricting the embeddings to concentrate on a pre-specified number of clusters. As a result, GCGQ can obtain more discriminative and clustering-friendly node representations that are consistent for different $k$s. This is considered to be an important advantage for real applications and data understanding. Extensive experiments show the superiority of GCGQ.

While GCGQ proves effective, it is not exempt from limitations. That is, the generality and efficiency of GCGQ are for different sought numbers of clusters $k$ on static data. Our future research will focus on improving the proposed quaternion representation learning for the adaptation of streaming data or even data with concept drift.

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

## A  EXPERIMENTAL SETTINGS

### A.1  DATASET SUMMARY

Table A.1: The statistics of ten graph datasets. $n$ is the number of nodes, $d$ is the dimension of attributes, and $k$ is the number of clusters provided by the labels of datasets.

| No. | Dataset | $n$ | $d$ | $k$ |
|---|---|---|---|---|
| 1 | ACM | 3025 | 1870 | 3 |
| 2 | WIKI | 2405 | 4973 | 17 |
| 3 | CITESEER | 3327 | 3703 | 6 |
| 4 | DBLP | 4057 | 334 | 4 |
| 5 | FILM | 7600 | 932 | 5 |
| 6 | CORNELL | 183 | 1703 | 5 |
| 7 | CORA | 2708 | 1433 | 7 |
| 8 | WISC | 251 | 1703 | 5 |
| 9 | UAT | 1190 | 239 | 4 |
| 10 | AMAP | 7650 | 745 | 8 |

Table A.1 shows the statistical summary of used attributed graph datasets.

### A.2  COUNTERPARTS AND GCGQ SETTINGS

In Comparison approaches, we follow their original settings. For traditional methods, the K-Means cluster the features without graph structure, and the Spectral clustering cluster the graph structure without feature matrix. They are executed 10 times for average scores. For conventional methods, we perform 200 epochs of unsupervised training of the GAE and VGAE, then use K-Means to cluster the generated embedding. For advanced and state-of-the-art clustering approaches, we reproduce their source code by following the original parameter setting in the source codes.

There are some hyper-parameters and settings of our method, i.e., the layer number, pre-training learning rate, pre-training iteration number, learning rate, iteration number, model regularization trade-off $\alpha$, and representation embedding loss trade-off $\beta$. We set Adam optimizer during experiments. The activation function of the graph encoder is ReLU for each layer. $\mathcal{L}_{reg}$ in the loss is the regularization of model, and L1 regularization is utilized. In the pre-training process, the hyper-parameter $\beta$ is set to 0.0001. For ten datasets, the neuron number of layer, the pre-training learning rate, pre-training iteration number, learning rate, and iteration number are set to $[512, 256, 128]$, $10^{-4}$, 10, $10^{-16}$, and 4, respectively. The $\alpha$ is set to $10^{-5}$ for CORA and DBLP, $5 \times 10^{-4}$ for CITESEER, FILM, and WISC, $10^{-6}$ for ACM, WIKI, AMAP, $10^{-4}$ for CORNELL and UAT. The $\beta$ is set to $2^{-10}$ for CORA, CORNELL, DBLP, WIKI, FILM, and CORNELL, $2^{-12}$ for CITESEER, $2^{-20}$ for ACM and UAT, 0 for AMAP.

### A.3  DESCRIPTION OF VALIDITY METRICS

We provide a more detailed description of validity metrics, which are Accuracy (ACC), Normalized Mutual Information (NMI), and Adjusted Rand Index (ARI) Zhou et al. (2022a).

ACC is a straightforward measure that calculates the percentage of correctly classified data points in the clustering results compared to ground truth. A higher accuracy indicates better performance. Given ground truth labels $Y = \{y_i | 1 \leq i \leq n\}$ and the predicted clusters $\hat{Y} = \{\hat{y}_i | 1 \leq i \leq n\}$, ACC is computed as

$$ACC(\hat{Y}, Y) = \max \frac{1}{n} \sum_{i=1}^{n} 1\{y_i = \hat{y}_i\}. \tag{1}$$

NMI quantifies the amount of shared information between two clusters. It ranges from 0 to 1, where 1 indicates perfect agreement and vice versa. Higher NMI values indicate better clustering

Table A.2: The Wilcoxon signed rank test with 95% confidence interval. The symbols "+" and "−" indicate the rejection and acceptance of the null hypothesis.

| Method | ACC | NMI | ARI |
|--------|-----|-----|-----|
| K-Means | + | + | + |
| Spectral-C | + | + | + |
| GAE | + | + | + |
| VGAE | + | + | + |
| ARGAE | + | + | + |
| ARVGAE | + | + | + |
| CONVERT | − | + | + |
| CCGC | + | + | + |
| DFCN | + | + | + |
| DAEGC | + | + | + |
| EGAE | + | + | + |

performance. The NMI can be computed by

$$NMI(\tilde{Y}, Y) = \frac{T(\tilde{Y}; Y)}{\frac{1}{2}\left[H(\tilde{Y}) + H(Y)\right]}, \tag{2}$$

where $H(Y)$ is entropy of $Y$ and $T(\tilde{Y}; Y)$ is mutual information between $\tilde{Y}$ and $Y$.

ARI measures the similarity between two clusters, taking into account both true positive and true negative matches while correcting for chance. It produces a value between -1 and 1. An ARI value close to 1 suggests strong agreement, close to 0 indicates random agreement, and negative values indicate disagreement. A higher ARI value indicates better clustering performance, and the ARI can be computed as

$$ARI = \frac{RI - \mathbb{E}(RI)}{\max(RI - \mathbb{E}(RI))}, \tag{3}$$

where

$$RI = \frac{TP + TN}{C_n^2}. \tag{4}$$

Here, $TP$ and $FP$ respectively denote the number of true positive pairs and true negative pairs, and $C_n^2$ is the number of possible object pairs.

## A.4 SETTINGS OF THE WILCOXON SIGNED-RANKS TEST

Here, we provide experimental settings of the Wilcoxon signed-ranks test for the results in Table A.2 of the submitted paper.

The Wilcoxon signed-ranks test is a non-parametric alternative to the paired t-test. It ranks the differences in performances of two classifiers for each dataset, ignoring the signs, and compares the ranks for the positive and the negative differences Demsar (2006). In general, the Wilcoxon signed-ranks test is used when we have paired data and try to observe if there is a significant change. If the test statistic is smaller than the critical value from a table (or if the p-value is below a chosen significance level), we can reject the null hypothesis, which suggests a significant difference between the paired data.

The procedures of the Wilcoxon signed-ranks test are: 1) Calculate the differences between paired observations. 2) Rank these differences in absolute rank values. 3) Assign positive or negative signs to the ranks based on the direction of the differences. 4) Sum the ranks of positive and negative differences separately. The smaller of the two sums is utilized for the test. If the smaller value is smaller than the critical value, we will reject the null hypothesis.

In our experiment, the Wilcoxon signed-ranks test is conducted to compare our method with other methods under different validity metrics on all the ten datasets. The procedures are as follows: 1) Formulate the hypothesis where the null hypothesis is that GCGQ does not exhibit a significant

---

**Algorithm 1** GCGQ: Graph Clustering based on Generalized QRL.

---

**Input**: Attributed graph $G = \{\mathbf{A}, \mathbf{X}\}$; Cluster number $k$; Loss weights $\alpha$ and $\beta$.
**Output**: $k$ non-overlapping sub-graphs $\{G_1, G_2, ..., G_k\}$.

1: Convert the adjacency matrix $\mathbf{A}$ into symmetric normalized Laplacian matrix $\tilde{\mathbf{A}}$;
2: **repeat**
3:     Project $\mathbf{X}$ into four views $\mathbf{F}_{\triangleright}$ by Eq. (1) and form a feature quaternion $\mathbf{F}$ as shown in Eq. (2);
4:     Encode $\mathbf{F}$ using quaternion graph encoders defined by Eqs. (3) and (4);
5:     Obtain the output embeddings $\mathbf{\Gamma}$ by the quaternion fusion operator defined in Eq. (5);
6:     Reconstruct the adjacency matrix $\hat{\mathbf{A}}$ from $\mathbf{\Gamma}$ according to Eq. (6);
7:     Compute the value of objective function $\mathcal{L}$ according to Eqs. (7) - (9);
8:     Update learnable parameters $\mathbf{W}_{\triangleright}^{L}, \mathbf{B}_{\triangleright}^{L}$ and $\mathbf{W}_{l}^{Q}$.
9: **until** maximum iterations reached
10: Perform spectral clustering to solve Eq. (10) based on $\hat{\mathbf{A}}$ reconstructed from the final $\mathbf{\Gamma}$.

---

difference, or perform equally, compared to other models under a specific validity metric. The alternative hypothesis is that GCGQ significantly outperforms other models. 2) Set the significance level at 0.01. 3) Calculate the p-value of the compared model performance. 4) Obtain the test results. If the p-value is less than the chosen significance level, we reject the null hypothesis, and vice versa, where a rejection suggests that GCGQ significantly outperforms the compared model.

## B  ALGORITHM AND COMPLEXITY ANALYSIS OF THE GCGQ

### B.1  ALGORITHM OF GCGQ

The algorithm process of GCGQ is shown in Algorithm 1.

### B.2  COMPUTATIONAL COMPLEXITY ANALYSIS

The time complexity of the proposed GQRL model is $\mathcal{O}(T[nd\hat{d} + n^2\hat{d}^2])$. We analyze it below. The training process of the model is composed of three parts: (1) quaternion projection, (2) quaternion graph convolution, and (3) graph reconstruction. For the quaternion projection, the dimensions of the input and projected features of each projector are $d$ and $\hat{d}$, respectively. Since four MLP layers are paralleled to project the attribute values of the $n$ nodes, the time complexity is thus $\mathcal{O}(4nd\hat{d})$. For the quaternion graph convolution, the feature quaternion in size $n \times (4 \times \hat{d})$ will be processed by $l$ stacked quaternion graph encoders. The parameters $\mathbf{W}_l^Q$ of each encoder are with the same scale as the feature quaternion. Hence, the time complexity of quaternion graph convolution is $\mathcal{O}(l(n4\hat{d})^2)$. For the graph reconstruction, the inner product is conducted on the matrix $\mathbf{\Gamma}$ with size $n \times \hat{d}$, which consumes $\mathcal{O}(n^2\hat{d})$. Assume the training of GQRL iterates $T$ times, the overall time complexity is $\mathcal{O}(T[4nd\hat{d} + l(n4\hat{d})^2 + n^2\hat{d}])$. By omitting the small constants and the terms with lower magnitude, the final complexity is nearly $\mathcal{O}(T[nd\hat{d} + n^2\hat{d}^2])$.

## C  COMPLEMENTARY EXPERIMENTAL RESULTS

### C.1  THE RESULTS OF WILCOXON SIGNED RANK TEST ON COMPARATIVE EXPERIMENTS

Table A.2 is the Wilcoxon signed rank test of comparative experiments results.

### C.2  BONFERRONI-DUNN TEST OF COMPARISON EXPERIMENT

In order to comprehensively demonstrate the superiority of our model compared to other methods, we conduct the Bonferroni-Dunn Test (BD test) Demsar (2006) based on the average rank (i.e., the 'AR' row) of the comparative experimental results in Table 1 of the main paper.

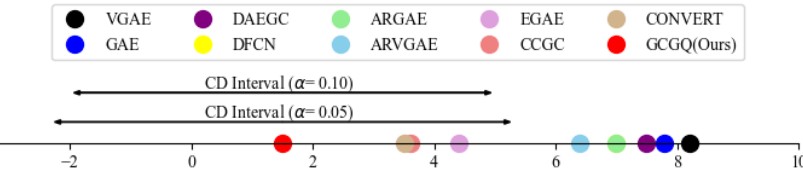

Figure A.1: Visualization of Bonferroni-Dunn (BD) test at confidence intervals 90% and 95%.

The Bonferroni-Dunn test is used to compare an algorithm with the remaining $k - 1$ counterparts. It involves comparing the differences in average ranks of various methods with a certain threshold value called Critical Difference (CD). The CD is defined as:

$$CD = q_\lambda \sqrt{\frac{p(p+1)}{6N}}, \tag{5}$$

where $q_\lambda$ is critical values for the BD test, $p$ is the number of compared methods, and $N$ is the number of dataset. If the rank difference between the two methods is higher than the CD, it indicates that the method with the higher average rank is statistically superior to the one with the lower average rank. Conversely, if the difference is lower than the CD, it suggests that there is no significant performance difference between the two methods.

Our BD test conduction procedures are as follows. 1) We obtain the ranks of the methods under all three validity metrics on all ten datasets. 2) The ranks under the three metrics are averaged to an overall rank of the corresponding method w.r.t. each certain dataset. 3) The average ranks on ten datasets are further averaged to an overall average rank of the methods, which are shown in Table 1 of the main paper.

According to Demsar (2006), we set the confidence intervals to 90% and 95%, and compute the CD by

$$CD_{0.10} = 3.4378, \tag{6}$$

and

$$CD_{0.05} = 3.7546, \tag{7}$$

where the $q_{0.10}$ and $q_{0.05}$ of ten classifiers are 2.539 and 2.773 according to Table 5(b) in reference Demsar (2006), the number of datasets $N$ is 10, and the number of compared methods $p$ is 10. Overall, it can be observed that GCGQ performs significantly better than the seven methods, as shown in Figure A.1.

## C.3 TRAINING CONVERGENCE EVALUATION

To demonstrate the convergence of our model, we show its convergence curves on all the ten benchmark datasets in Table A.2.

The overall trend of the loss convergence curves indicates a steady decrease in loss, which suggests that the model can effectively learn from the training data. Although there are minor fluctuations in the loss curves on some datasets, the loss decreasing tends stable when approaching the pre-set 50 epoch of training. In summary, the training convergence evaluation illustrates that our model can be effectively trained for learning representation and clustering.

## C.4 SENSITIVITY EVALUATION OF HYPER-PARAMETERS

The sensitivity of GCGQ to the trade-off hyper-parameters $\alpha$ and $\beta$ is evaluated on the datasets as shown in Figure A.3. Note that when evaluating sensitivity to one parameter, another one is fixed at the corresponding settings in Appendix A.2. From the results, it is not surprising that a too-large value of $\alpha$ or $\beta$ leads to generating objective biased representations such that GCGQ obtains undesired clustering performance. The results also confirmed that GCGQ is insensitive to $\alpha$ and $\beta$ in the value range around the parameter settings adopted for the aforementioned experiments.

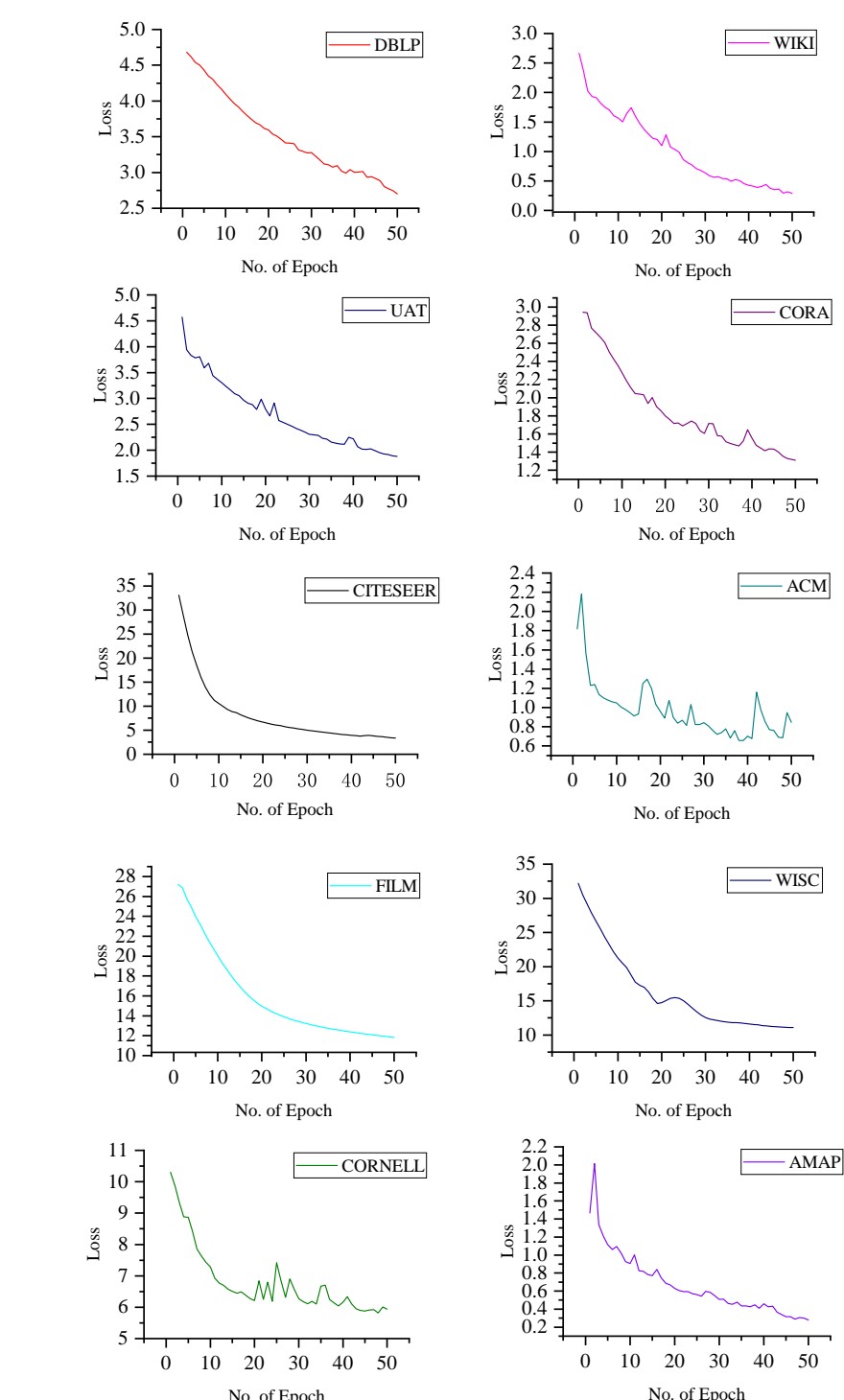

Figure A.2: Convergence curves of the GCGQ on ten datasets.

## C.5 VISUAL RESULTS

The supplementary $t$-SNE visualization results of the representations learned by different methods on the ACM and DBLP datasets are shown in Fig A.4.

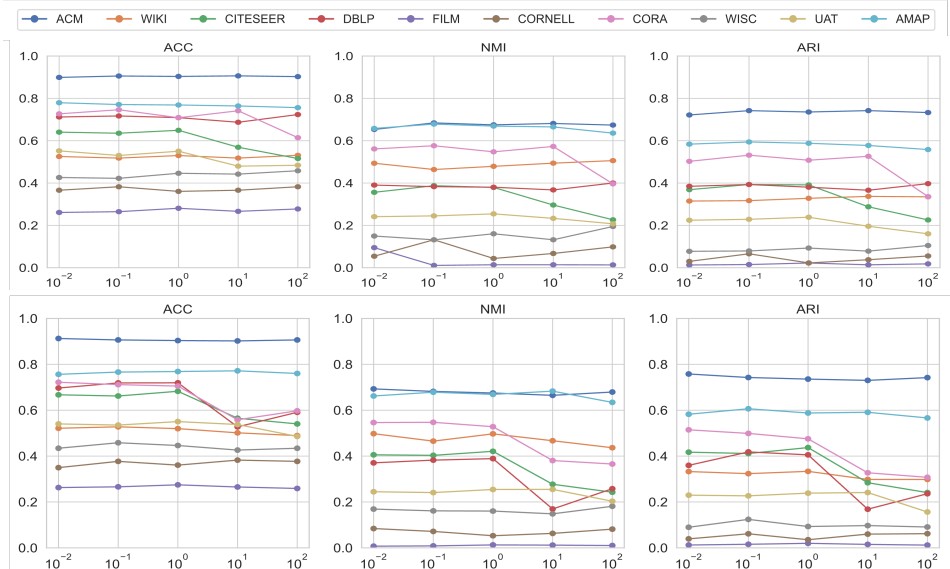

Figure A.3: Sensitivity analysis of the trade-off parameters of the loss terms, i.e., $\alpha$ for $\mathcal{L}_{reg}$ (the upper row) and $\beta$ for $\mathcal{L}_{sc}$ (the lower row), on all the ten datasets (marked in lines with different colors). $x$-axes indicate the values of $\alpha$ and $\beta$.

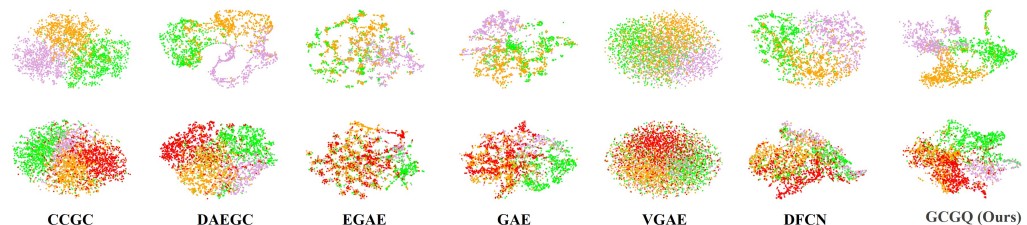

Figure A.4: $t$-SNE visualization on ACM datasets. The first and second rows correspond to ACM and DBLP, respectively.

To intuitively compare the ablated versions of GCGQ, the representations learned by them and GCGQ are also compared using $t$-SNE on the CORA dataset in Figure A.5.

For all the visualization results in this section, the observations and conclusions are consistent with the corresponding results in the main paper, so we do not provide redundant discussions here.

## D  DISCUSSION ABOUT REMARK AND PROOF

### D.1  DETAILED REMARK OF LEARNING ABILITY

We provide a more detailed analysis of "Remark 1" in Section 3.2 of the submitted paper. The more detailed Remark 1 is given below.

**Remark 1** *Degree of Freedom. According to Eq. (3) in main paper, learnable parameters in our model, i.e., $\mathbf{W}_{\mathbb{H}}^{Q} = \{\mathbf{W}_r^Q, \mathbf{W}_x^Q, \mathbf{W}_y^Q, \mathbf{W}_z^Q\}$, yields 16 pairs of feature interaction. In contrast, realizing the same scale interaction in real-value space requires 4 times of parameters. This illustrates the learning efficiency of the proposed model. Detailed analysis is given below.*

*Given model input*

$$\mathbf{F} = \{\mathbf{F}_r, \mathbf{F}_x, \mathbf{F}_y, \mathbf{F}_z\}, \tag{8}$$

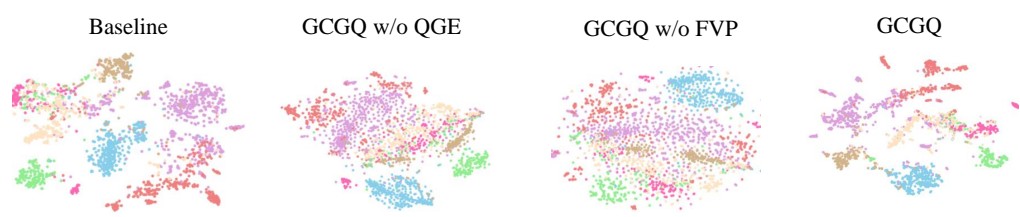

Figure A.5: $t$-SNE visualization of the ablated variants of GCGQ on CORA dataset.

where $\mathbf{F} \in \mathbb{H}^{n \times (4 \times \hat{d})}$, $\hat{d}$ *indicates the dimension of input. Then, we define the learnable parameters of quaternion representation as* $\mathbf{W}_{\mathbb{H}}^{Q} \in \mathbb{H}^{(4 \times \hat{d}) \times (4 \times \tilde{d})}$, *which contains four part of parameters* $\{\mathbf{W}_r^Q, \mathbf{W}_x^Q, \mathbf{W}_y^Q, \mathbf{W}_z^Q\}$, *where* $\tilde{d}$ *is the output dimension, and* $\mathbf{W}_i^Q \in \mathbb{H}^{\hat{d} \times \tilde{d}}$ *with* $i \in \{r, x, y, z\}$.

*According to the Hamilton product in the quaternion system, the learnable parameters let the features in* $\mathbf{F}$ *interact by*

$$
\begin{aligned}
\mathbf{F}^Q = \mathbf{F} \otimes \mathbf{W}^Q = \ &\mathbf{W}_r^Q \mathbf{F}_r - \mathbf{W}_x^Q \mathbf{F}_x - \mathbf{W}_y^Q \mathbf{F}_y - \mathbf{W}_z^Q \mathbf{F}_z \\
&+ \mathbf{W}_x^Q \mathbf{F}_r + \mathbf{W}_r^Q \mathbf{F}_x - \mathbf{W}_z^Q \mathbf{F}_y + \mathbf{W}_y^Q \mathbf{F}_z \\
&+ \mathbf{W}_y^Q \mathbf{F}_r + \mathbf{W}_z^Q \mathbf{F}_x + \mathbf{W}_r^Q \mathbf{F}_y - \mathbf{W}_x^Q \mathbf{F}_z \\
&+ \mathbf{W}_z^Q \mathbf{F}_r - \mathbf{W}_y^Q \mathbf{F}_x + \mathbf{W}_x^Q \mathbf{F}_y + \mathbf{W}_r^Q \mathbf{F}_z
\end{aligned}
\tag{9}
$$

*where* $\mathbf{F}^Q \in \mathbb{H}^{n \times (4 \times \tilde{d})}$. *It is intuitive that such an operation yields learning with a 16-Degree of Freedom (DoF).*

*In the following, we design a real-value model with the same DoF, and observe how many parameters are required for comparison. For intuitive comparison, we define the parameters of the real-value model in a similar form as that of the quaternion model, i.e.,* $\mathbf{W}_i^R \in \mathbb{R}^{4 \times (\hat{d} \times \tilde{d})}$ *with* $i \in \{r, x, y, z\}$. *The superscript $R$ indicates that these are the parameters of the real-value model. Accordingly, all the features in* $\mathbf{F}$ *interact through the parameters by*

$$
\mathbf{F}^R = [\mathbf{F}_r \mathbf{W}_r^R, \mathbf{F}_x \mathbf{W}_x^R, \mathbf{F}_y \mathbf{W}_y^R, \mathbf{F}_z \mathbf{W}_z^R],
\tag{10}
$$

*where* $\mathbf{F}^R \in \mathbb{R}^{n \times (4 \times \tilde{d})}$ *is the output matrix, and* $\mathbf{W}_r^R, \mathbf{W}_x^R, \mathbf{W}_y^R, \mathbf{W}_z^R$ *can be written as*

$$
\begin{aligned}
\mathbf{W}_r^R &= [\mathbf{W}_1^R, \mathbf{W}_2^R, \mathbf{W}_3^R, \mathbf{W}_4^R] \\
\mathbf{W}_x^R &= [\mathbf{W}_5^R, \mathbf{W}_6^R, \mathbf{W}_7^R, \mathbf{W}_8^R] \\
\mathbf{W}_y^R &= [\mathbf{W}_9^R, \mathbf{W}_{10}^R, \mathbf{W}_{11}^R, \mathbf{W}_{12}^R] \\
\mathbf{W}_z^R &= [\mathbf{W}_{13}^R, \mathbf{W}_{14}^R, \mathbf{W}_{15}^R, \mathbf{W}_{16}^R].
\end{aligned}
\tag{11}
$$

*Accordingly, the output feature* $\mathbf{F}^R$ *is written as*

$$
\mathbf{F}^R = \begin{bmatrix} \mathbf{W}_1^R \mathbf{F}_r + \mathbf{W}_2^R \mathbf{F}_r + \mathbf{W}_3^R \mathbf{F}_r + \mathbf{W}_4^R \mathbf{F}_r \\ \mathbf{W}_5^R \mathbf{F}_x + \mathbf{W}_6^R \mathbf{F}_x + \mathbf{W}_7^R \mathbf{F}_x + \mathbf{W}_8^R \mathbf{F}_x \\ \mathbf{W}_9^R \mathbf{F}_y + \mathbf{W}_{10}^R \mathbf{F}_y + \mathbf{W}_{11}^R \mathbf{F}_y + \mathbf{W}_{12}^R \mathbf{F}_y \\ \mathbf{W}_{13}^R \mathbf{F}_z + \mathbf{W}_{14}^R \mathbf{F}_z + \mathbf{W}_{15}^R \mathbf{F}_z + \mathbf{W}_{16}^R \mathbf{F}_z \end{bmatrix}^{\top}.
\tag{12}
$$

*Obviously, there are 16 parameter matrices that are of the same size of* $\mathbf{W}_i^R \in \mathbb{R}^{4 \times (\hat{d} \times \tilde{d})}$, *and it can be concluded that if a real-value model is adopted to realize the same DoF as that of the quaternion-value model, a four-time model parameters scale will be involved. In other words, with the same number of parameters, the quaternion-value model can achieve a higher DoF than real-value models for more informative representation and cluster learning.*

## D.2 Proof of Degree of Freedom

According to above intuitive discussions, the learnable parameters $\mathbf{W}^{\mathbb{H}} = \{\mathbf{W}_r^{\mathbb{H}}, \mathbf{W}_x^{\mathbb{H}}, \mathbf{W}_y^{\mathbb{H}}, \mathbf{W}_z^{\mathbb{H}}\}$ can generate 16 features learning pairs, which realize the same ability of 16 Degree of Freedom

(DoF) feature learning in real-value field. In the same DoF representation, the number of real-value parameters is $4\times$ of the quaternion-value model. Thus, with the same number of parameters, the quaternion-value model can achieve a higher DoF that helps to explore features for more informative representation and cluster learning.

Now, we prove the DoF in mathematics. We follow the parameter initialization method in Parcollet et al. (2019), and use $W$ instead of $\mathbf{W}^{\mathbb{H}}$ to prove the Degree of Freedom (DoF) of GCGQ here. The initialization equations are derived from the polar form of a weight $w$ of $W$, and $w$ has a polar form defined as:

$$w = |w|e^{q_{img}^{\triangleleft}\theta} = |w|(cos(\theta) + q_{img}^{\triangleleft}sin(\theta)), \tag{13}$$

with $q_{\text{img}}^{\triangleleft} = 0 + x\mathbf{i} + y\mathbf{j} + z\mathbf{k}$ a purely imaginary and normalized quaternion. Therefore, $w$ can be computed following:

$$\begin{aligned}
w_{\mathbf{r}} &= \varphi\cos(\theta), \\
w_{\mathbf{i}} &= \varphi q_{\text{img-i}}^{\triangleleft}\sin(\theta), \\
w_{\mathbf{j}} &= \varphi q_{\text{img-j}}^{\triangleleft}\sin(\theta), \\
w_{\mathbf{k}} &= \varphi q_{\text{img-k}}^{\triangleleft}\sin(\theta).
\end{aligned} \tag{14}$$

However, $\varphi$ represents a randomly generated variable with respect to the variance of the quaternion weight and the selected initialization criterion. The initialization process follows Glorot & Bengio (2010) and He et al. (2015) to derive the variance of the quaternion-valued weight parameters. Indeed, the variance of $\mathbf{W}$ has to be investigated:

$$\text{Var}(W) = \mathbb{E}\left(|W|^2\right) - [\mathbb{E}(|W|)]^2, \tag{15}$$

$[\mathbb{E}(|W|)]^2$ equals to 0 since the weight distribution is symmetric around 0. Nonetheless, the value of $\text{Var}(W) = \mathbb{E}\left(|W|^2\right)$ is not trivial in the case of quaternion-valued matrices. Indeed, $W$ follows a Chi-distribution with four degrees of freedom (DoFs) and $\mathbb{E}\left(|W|^2\right)$ is expressed and computed as follows:

$$\mathbb{E}\left(|W|^2\right) = \int_0^\infty x^2 f(x)\mathrm{d}x. \tag{16}$$

With $f(x)$ is the probability density function with four DoFs. A four-dimensional vector $X = \{A, B, C, D\}$ is considered to evaluate the density function $f(x)$. $X$ has components that are normally distributed, centered at zero, and independent. Then, $A, B, C$ and $D$ have density functions:

$$f_A(x;\sigma) = f_B(x;\sigma) = f_C(x;\sigma) = f_D(x;\sigma) = \frac{e^{-x^2/2\sigma^2}}{\sqrt{2\pi\sigma^2}}. \tag{17}$$

The four-dimensional vector $X$ has a length $L$ defined as $L = \sqrt{A^2 + B^2 + C^2 + D^2}$ with a cumulative distribution function $F_L(x;\sigma)$ in the 4-sphere (n-sphere with $n = 4$) $S_x$ :

$$F_L(x;\sigma) = \iiint \int_{S_x} f_A(x;\sigma)f_B(x;\sigma)f_C(x;\sigma)f_D(x;\sigma)\mathrm{d}S_x, \tag{18}$$

where $S_x = \left\{(a, b, c, d) : \sqrt{a^2 + b^2 + c^2 + d^2} < x\right\}$ and $\mathrm{d}S_x = \mathrm{d}a\,\mathrm{d}b\,\mathrm{d}c\,\mathrm{d}d$. The polar representations of the coordinates of $X$ in a 4-dimensional space are defined to compute $\mathrm{d}S_x$ :

$$\begin{aligned}
a &= \rho\cos\theta \\
b &= \rho\sin\theta\cos\phi \\
c &= \rho\sin\theta\sin\phi\cos\psi \\
d &= \rho\sin\theta\sin\phi\sin\psi,
\end{aligned} \tag{19}$$

where $\rho$ is the magnitude ( $\rho = \sqrt{a^2 + b^2 + c^2 + d^2}$ ) and $\theta$, $\phi$, and $\psi$ are the phases with $0 \le \theta \le \pi$, $0 \le \phi \le \pi$ and $0 \le \psi \le 2\pi$. Then, $\mathrm{d}S_x$ is evaluated with the Jacobian $J_f$ of $f$ defined as:

$$
J_f = \frac{\partial(a, b, c, d)}{\partial(\rho, \theta, \phi, \psi)} = \frac{\mathrm{d}a\,\mathrm{d}b\,\mathrm{d}c\,\mathrm{d}d}{\mathrm{d}\rho\mathrm{d}\theta\mathrm{d}\phi\mathrm{d}\psi} = \begin{vmatrix} \frac{\mathrm{d}a}{\mathrm{d}\rho} & \frac{\mathrm{d}a}{\mathrm{d}\theta} & \frac{\mathrm{d}a}{\mathrm{d}\phi} & \frac{\mathrm{d}a}{\mathrm{d}\psi} \\ \frac{\mathrm{d}b}{\mathrm{d}\rho} & \frac{\mathrm{d}b}{\mathrm{d}\theta} & \frac{\mathrm{d}b}{\mathrm{d}\phi} & \frac{\mathrm{d}b}{\mathrm{d}\psi} \\ \frac{\mathrm{d}c}{\mathrm{d}\rho} & \frac{\mathrm{d}c}{\mathrm{d}\theta} & \frac{\mathrm{d}c}{\mathrm{d}\phi} & \frac{\mathrm{d}c}{\mathrm{d}\psi} \\ \frac{\mathrm{d}d}{\mathrm{d}\rho} & \frac{\mathrm{d}d}{\mathrm{d}\theta} & \frac{\mathrm{d}d}{\mathrm{d}\phi} & \frac{\mathrm{d}d}{\mathrm{d}\psi} \end{vmatrix}
\tag{20}
$$

$$
= \begin{vmatrix} \cos\theta & -\rho\sin\theta & 0 & 0 \\ \sin\theta\cos\phi & \rho\sin\theta\cos\phi & -\rho\sin\theta\sin\phi & 0 \\ \sin\theta\sin\phi\cos\psi & \rho\cos\theta\sin\phi\cos\psi & \rho\sin\theta\cos\phi\cos\psi & -\rho\sin\theta\sin\phi\sin\psi \\ \sin\theta\sin\phi\sin\psi & \rho\cos\theta\sin\phi\sin\psi & \rho\sin\theta\cos\phi\sin\psi & \rho\sin\theta\sin\phi\cos\psi \end{vmatrix}.
$$

And,

$$
J_f = \rho^3 \sin^2\theta \sin\phi.
\tag{21}
$$

Therefore, by the Jacobian $J_f$, we have the polar form:

$$
\mathrm{d}a\,\mathrm{d}b\,\mathrm{d}c\,\mathrm{d}d = \rho^3 \sin^2\theta \sin\phi \mathrm{d}\rho\mathrm{d}\theta\mathrm{d}\phi\mathrm{d}\psi.
\tag{22}
$$

Then, writing Eq. (18) in polar coordinates, we obtain:

$$
\begin{aligned}
F_L(x, \sigma) &= \left(\frac{1}{\sqrt{2\pi\sigma^2}}\right)^4 \iiint \int_0^x e^{-a^2/2\sigma^2} e^{-b^2/2\sigma^2} e^{-c^2/2\sigma^2} e^{-d^2/2\sigma^2} \; \mathrm{d}S_x \\
&= \frac{1}{4\pi^2\sigma^4} \int_0^{2\pi} \int_0^{\pi} \int_0^{\pi} \int_0^x e^{-\rho^2/2\sigma^2} \rho^3 \sin^2\theta \sin\phi \mathrm{d}\rho\mathrm{d}\theta\mathrm{d}\phi\mathrm{d}\psi \\
&= \frac{1}{4\pi^2\sigma^4} \int_0^{2\pi} \mathrm{d}\psi \int_0^{\pi} \sin\phi\mathrm{d}\phi \int_0^{\pi} \sin^2\theta\mathrm{d}\theta \int_0^x \rho^3 e^{-\rho^2/2\sigma^2} \; \mathrm{d}\rho \\
&= \frac{1}{4\pi^2\sigma^4} 2\pi 2 \left[\frac{\theta}{2} - \frac{\sin 2\theta}{4}\right]_0^{\pi} \int_0^x \rho^3 e^{-\rho^2/2\sigma^2} \; \mathrm{d}\rho \\
&= \frac{1}{4\pi^2\sigma^4} 4\pi \frac{\pi}{2} \int_0^x \rho^3 e^{-\rho^2/2\sigma^2} \; \mathrm{d}\rho
\end{aligned}
\tag{23}
$$

Then,

$$
F_L(x, \sigma) = \frac{1}{2\sigma^4} \int_0^x \rho^3 e^{-\rho^2/2\sigma^2} \; \mathrm{d}\rho.
\tag{24}
$$

The probability density function for $X$ is the derivative of its cumulative distribution function, which by the fundamental theorem of calculus is:

$$
\begin{aligned}
f_L(x, \sigma) &= \frac{\mathrm{d}}{\mathrm{d}x} F_L(x, \sigma) \\
&= \frac{1}{2\sigma^4} x^3 e^{-x^2/2\sigma^2}
\end{aligned}
\tag{25}
$$

The expectation of the squared magnitude becomes:

$$
\begin{aligned}
\mathbb{E}\left(|W|^2\right) &= \int_0^{\infty} x^2 f(x)\mathrm{d}x \\
&= \int_0^{\infty} x^2 \frac{1}{2\sigma^4} x^3 e^{-x^2/2\sigma^2} \; \mathrm{d}x \\
&= \frac{1}{2\sigma^4} \int_0^{\infty} x^5 e^{-x^2/2\sigma^2} \; \mathrm{d}x
\end{aligned}
\tag{26}
$$

With integration by parts we obtain:

$$
\begin{aligned}
\mathbb{E}\left(|W|^2\right) &= \frac{1}{2\sigma^4} \left(- x^4 \sigma^2 e^{-x^2/2\sigma^2}\Big|_0^{\infty} + \int_0^{\infty} \sigma^2 4x^3 e^{-x^2/2\sigma^2} \; \mathrm{d}x\right) \\
&= \frac{1}{2\sigma^2} \left(- x^4 e^{-x^2/2\sigma^2}\Big|_0^{\infty} + \int_0^{\infty} 4x^3 e^{-x^2/2\sigma^2} \; \mathrm{d}x\right)
\end{aligned}
\tag{27}
$$

The expectation $\mathbb{E}\left(|W|^2\right)$ is the sum of two terms. The first one:

$$
\begin{aligned}
- x^4 e^{-x^2/2\sigma^2} \Big|_0^\infty &= \lim_{x \to +\infty} -x^4 e^{-x^2/2\sigma^2} - \lim_{x \to +0} x^4 e^{-x^2/2\sigma^2} \\
&= \lim_{x \to +\infty} -x^4 e^{-x^2/2\sigma^2}
\end{aligned}
\tag{28}
$$

Based on the L'Hôopital's rule, the undetermined limit becomes:

$$
\begin{aligned}
\lim_{x \to +\infty} -x^4 e^{-x^2/2\sigma^2} &= \lim_{x \to +\infty} \frac{x^4}{e^{x^2/2\sigma^2}} \\
&= \lim_{x \to +\infty} \frac{1}{(24/\sigma^2)(P(x)e^{x^2/2\sigma^2})} \\
&= 0.
\end{aligned}
\tag{29}
$$

With $P(x)$ is polynomial and has a limit to $+\infty$. The second term is calculated in a same way (integration by parts) and $\mathbb{E}(|W|^2)$ becomes from Eq. (27):

$$
\begin{aligned}
\mathbb{E}(|W|^2 &= \frac{1}{2\sigma^2} \int_0^\infty 4x^3 e^{-x^2/2\sigma^2} \, \mathrm{d}x \\
&= \frac{2}{\sigma^2} \left( x^2 \sigma^2 e^{-x^2/2\sigma^2} \Big|_0^\infty + \int_0^\infty \sigma^2 2x e^{-x^2/2\sigma^2} \, \mathrm{d}x \right)
\end{aligned}
\tag{30}
$$

The limit of first term is equals to 0 with the same method than in Eq. (29). Therefore, the expectation is:

$$
\begin{aligned}
\mathbb{E}(|W|^2) &= 4 \left( \int_0^\infty x e^{-x^2/2\sigma^2} \, \mathrm{d}x \right) \\
&= 4\sigma^2.
\end{aligned}
\tag{31}
$$

And finally, the variance is:

$$
Var(|W|) = 4\sigma^2.
\tag{32}
$$

This proof demonstrates that the DoF of quaternion weights in the encoders is four times higher than the weights in the conventional graph encoders.

