# OpenReview forum: "Attributed Graph Clustering via Generalized Quaternion Representation Learning"
_ICLR.cc/2025/Conference — ICLR 2025 Conference Withdrawn Submission_

### Official Review · Reviewer_VdUL · 2024-10-28

**Soundness:** 2
**Presentation:** 3
**Contribution:** 2
**Rating:** 5
**Confidence:** 4

**Summary:**

The paper introduces a novel approach for attributed graph clustering. The proposed GCGQ utilizes quaternion operations within a graph auto-encoder framework to address the over-smoothing effect commonly encountered in GCNs. The model aims to learn discriminative node representations that are robust to varying cluster granularities, without the need to pre-specify the number of clusters.

**Strengths:**

1. The paper presents an approach on graph clustering by incorporating quaternion operations, which is a mathematical concept that extends the capabilities of traditional graph encoding methods.
2. GCGQ doesn't require a predetermined number of clusters, making it more adaptable to different data sets and scenarios where the target number of clusters may not be known beforehand.

**Weaknesses:**

1. The motivation for introducing quaternion operations by the authors doesn’t seem very natural. Is it simply because previous methods were insufficient in modeling attribute features, leading to the introduction of this mechanism? Why not consider many conventional techniques for incorporating attribute features?
2. Could the authors provide more insight into how quaternion encoding impacts the learning process and the quality of the resulting clusters?
3. Does the authors conduct experiments to demonstrate that the proposed method addresses the over-smoothing problem in GNNs?
4. What is the sensitivity of the model's performance to changes in the hyperparameters, especially the trade-off parameters α and β?
5. Can the authors discuss the scalability of their model, particularly for very large graphs with millions of nodes?
6. The authors' baseline methods lack more recent works, especially those from 2024.

**Questions:**

See above.

---

> ### Comment · Reviewer_VdUL · 2024-11-27
>
> The author did not respond the reviewers' comments, so I will keep my score.

---

### Official Review · Reviewer_vSph · 2024-10-31

**Soundness:** 3
**Presentation:** 2
**Contribution:** 3
**Rating:** 6
**Confidence:** 4

**Summary:**

This paper proposes a generalized quaternion representation learning method for clustering with attribute maps, aiming to solve the common “oversmoothing” and “over-dominance” problems in GCN. The authors introduce quaternion operations into the encoder, which can efficiently learn structured feature representations without deepening the network through multi-view projection and quaternion map convolution. In addition, the paper designs a generalized graph clustering objective that avoids the problem of pre-specifying the number of clusters k in traditional clustering methods. Experimental results show that the proposed method achieves better performance than existing methods on multiple benchmark datasets.

**Strengths:**

1. The paper presents an innovative approach to introduce quaternion operations into graph convolutional networks, which is the first application in unsupervised learning tasks on graph data.

2. The over-smoothing problem in GCN is successfully overcome by multi-view projection and shallow network, and the graph attribute information is also preserved.

3. The proposed generalized clustering objective can be adapted to different numbers of clusters k, which avoids the need for repeated training of the model in traditional methods and enhances the flexibility in practical applications.

**Weaknesses:**

1. Some parts of the paper have complex mathematical derivations and symbolic definitions that are difficult to read. It is recommended that more visual illustrations and examples be added to help understand the quaternionic convolution module and clustering process.

2. The limitations of the quaternion representation or its performance differences with existing methods are not discussed in depth, especially in terms of computational resources or time overheads.

**Questions:**

1. How does the introduction of quaternionic convolution specifically improve the interaction between different dimensional properties? Can clearer experiments be provided to demonstrate its effect?

2. Does quaternion representation have higher demand on computational resources in practical applications? In particular, is there a significant time and memory overhead problem when dealing with large-scale graph data compared to other methods?

---

### Official Review · Reviewer_RcFz · 2024-11-03

**Soundness:** 2
**Presentation:** 2
**Contribution:** 1
**Rating:** 1
**Confidence:** 5

**Summary:**

The paper showcases a method to perform clustering on attributed graphs by using quaternion learning without having to specify the cluster size while training.
The method involves splitting the input into four coupled branches (a quaternion field $F_r,F_x,F_y,F_z$) using different encoders ($W_r,W_x,W_y,W_z$), then performing an efficient Hamiltonian product in the hyper-complex quaternion space, and finally performing a "quaternion fusion" operation (taking the average of the four quaternion parts) to get an embedding matrix $\Gamma$. This is then passed through a simple inner-product decoder to obtain the reconstructed adjacency $\hat{A} = \Gamma\cdot\Gamma^T$. Standard reconstruction loss is calculated for the adjacency, and a spectral clustering term $\text{Tr}(\Gamma^TL\Gamma)$, where $L = D - \hat{A}$, the Laplacian.
This completes the training stage and starts the explicit clustering stage. Now, we get an $n \times k$ matrix $E$ as the top-$k$ eigenvectors of the Laplacian which are considered as the node representations which have strong cluster discriminability, and are then passed to K-Means to do the clustering ($k$ is number of clusters, $n$ is number of nodes).
The method makes sense and the experiments support the same, although a comparison is not made to recent methods (this is detailed in the Weaknesses and Questions sections).

**Strengths:**

The authors conducted a comprehensive experimental validation across six datasets, utilizing three widely recognized metrics: NMI, ARI, and Accuracy. The inclusion of the Wilcoxon Signed Ranks Test, along with the computation of average performance ranks, are interesting additions to the paper.
- The complexity analysis of the GCGQ method is a good addition that gives depth to the study.
- The visualizations are well made.

**Weaknesses:**

The current submission appears to lack sufficient technical novelty. The proposed method closely mirrors an approach presented in an already accepted paper [^1], presenting only minor variations or subsets of existing work without significant new contributions. The referenced paper involves clustering non-graph data by constructing a graph, applying Graph Quaternion Learning, and then performing spectral clustering followed by K-means. Notably, both papers share similar diagrams, explanations, and even training specifics, underscoring the overlap in methodology without meaningful advancement in the proposed work.


I strongly recommend that the authors refer to the other paper, as it directly addresses similar challenges with a comparable approach. It is essential to clarify how your work diverges from or improves upon this existing method. Merely replicating or making minor adjustments to an established approach does not demonstrate significant innovation.

At its current stage, the contribution of the paper seems insufficient for publication. I encourage the authors to revisit the related work, particularly the referenced paper, and provide a more robust justification of their contributions.

[^1]: Junyang Chen, Yiu-ming Cheung. "QGRL: Quaternion Graph Representation Learning for Heterogeneous Feature Data Clustering." _Proceedings of the 30th SIGKDD Conference on Knowledge Discovery and Data Mining_. 2024.

**Questions:**

- There are some notational inconsistencies in the paper: In Section 3.1, matrix L = I + A (self-loop adjacency matrix), but in Section 3.3 it is redefined as the Laplacian Matrix.
- The method is referenced by multiple names including "GCGQ" and "GQRL" including the appendix.
- The baselines used for comparison are relatively old. Newer methods which use contrastive learning are achieving much higher performance on standard datasets like Cora, Citeseer and Pubmed. It is important to include the following papers as baselines: [^2], [^3], [^4], [^5]
- All the graph datasets used in the paper have a few thousand nodes. Was the reason for not experimenting on large graph datasets that the complexity of the method is poor $\approx O(n^2\hat{d}^2)$? It might be useful to add results for a few large datasets to the paper.
- A few recent papers ([^3], [^4], [^5]) which offer superior performance on Cora and Citeseer have complexities $O(n^2k + n\hat{d}k)$ lower than the GCQC method. It is important to address the advantages of GCQC over these existing methods.
- The paper talks about a "over-dominating" effect separately from the "over-smoothing" effect. However, they appear to originate from the same thing. The paper states "That is, the embeddings of topology-adjacent but attribute-dissimilar nodes will be similar due to the information aggregation dominated by the graph topology." which is what over-smoothing is.
- There are multiple typing errors and similar mistakes in the paper
	- Typo in figure 1: convolusion -> convolution
	- In line 228, the authors write $W^Q \in H^{n\times(4\hat{d})}$, which is a typo and should be $4d\times4\hat{d}$ so that the Hamiltonian product b/w $F$ and $W^Q$ can succeed.
	- In eqn. 5, the method involves a "quarternion fusion" operator, which the authors clarify to be just an average of the four parts. Why is this chosen to be the operator of choice and how do other operators such as min/max change the behaviour of the method? This seems to be analogous to the pooling operation in many ways, can any type of pooling work here?
	- In line 285, the authors write "learned embeddings of the node
	attributes indicated by $\hat{A}$", which is a typo. The correct definition is given in line 291.

[^2] N. Mrabah, M. Bouguessa, and R. Ksantini, ‘Escaping Feature Twist: A Variational Graph Auto-Encoder for Node Clustering’, in Proceedings of the Thirty-First International Joint Conference on Artificial Intelligence, IJCAI-22, 7 2022, pp. 3351–3357.
[^3] Yue Liu, Xihong Yang, Sihang Zhou, Xinwang Liu, Zhen Wang, Ke Liang, Wenxuan Tu, Liang Li, Jingcan Duan, and Cancan Chen. 2023. Hard sample aware network for contrastive deep graph clustering. In Proceedings of the Thirty-Seventh AAAI Conference on Artificial Intelligence and Thirty-Fifth Conference on Innovative Applications of Artificial Intelligence and Thirteenth Symposium on Educational Advances in Artificial Intelligence (AAAI'23/IAAI'23/EAAI'23), Vol. 37. AAAI Press, Article 1002, 8914–8922. https://doi.org/10.1609/aaai.v37i7.26071
[^4] F. Devvrit, A. Sinha, I. Dhillon, and P. Jain, ‘S3GC: Scalable Self-Supervised Graph Clustering’, in Advances in Neural Information Processing Systems, 2022, vol. 35, pp. 3248–3261.
[^5] Nairouz Mrabah, Mohamed Bouguessa, Mohamed Fawzi Touati, and Riadh Ksantini. Rethinking graph autoencoder models for attributed graph clustering. IEEE Transactions on Knowledge and Data Engineering, pp. 1–15, 2022. doi: 10.1109/TKDE.2022.3220948

**Details Of Ethics Concerns:**

The current submission appears to lack sufficient technical novelty. The proposed method closely mirrors an approach presented in an already accepted paper [^1], presenting only minor variations or subsets of existing work without significant new contributions. The referenced paper involves clustering non-graph data by constructing a graph, applying Graph Quaternion Learning, and then performing spectral clustering followed by K-means. Notably, both papers share similar diagrams, explanations, and even training specifics, underscoring the overlap in methodology without meaningful advancement in the proposed work.

[^1]: Junyang Chen, Yiu-ming Cheung. "QGRL: Quaternion Graph Representation Learning for Heterogeneous Feature Data Clustering." _Proceedings of the 30th SIGKDD Conference on Knowledge Discovery and Data Mining_. 2024.

---

### Official Review · Reviewer_BLf5 · 2024-11-03

**Soundness:** 1
**Presentation:** 2
**Contribution:** 2
**Rating:** 3
**Confidence:** 4

**Summary:**

This paper introduces quaternion operations into attributed graph clustering. Specifically, this paper proposes Four-View Projection (FVP) to project original node attributes to four views that satisfy the needs of applying quaternion operations. Further, this paper proposes Quaternion Graph Encoders (QGE) that incorporates structural knowledge during representation learning. Finally, this paper presents a clustering-friendly loss that does not need a pre-specified number of clusters.

**Strengths:**

S1: Observation of the “over-dominating” phenomenon.

S2: Introduction of quaternion operations into attributed graph clustering, which significantly boosts learning capacity.

S3: Strong performance across multiple datasets and evaluation metrics.

**Weaknesses:**

W1: Several important claims are not well supported.

W2: The rationale for adopting quaternion operations is not thoroughly validated.

W3: Ablation studies are not comprehensive enough.

**Questions:**

Q1: The authors state that conventional GCN-based methods struggle to capture global structural information due to the over-smoothing issue associated with deep GCN encoders. However, it appears that the proposed GCGQ may also be vulnerable to this issue, as it also depends on stacking graph convolutional layers to extract high-order structural information.

Q2: Additional empirical or theoretical evidence is required to support the claim that GCGQ effectively mitigates the over-dominating issue.

Q3: The rationale for adopting quaternion operations is not thoroughly validated. While adopting quaternion operations may enhance learning capacity at the same parameter size (higher DoF), the actual benefits for the attributed graph clustering task are insufficiently demonstrated empirically.

Q4: The ablation study should include a model variant that omits the clustering-oriented loss (L_sc) to evaluate its effectiveness. The contribution of this loss to the clustering task is critical; if it proves ineffective, the paper's distinction from conventional graph representation learning methods, which also do not necessitate a predetermined number of clusters, would diminish.

Q5: The citation format within the main text requires correction. The use of \citet should be limited to places where inline references are desired.

---

### Note · Authors · 2024-12-17

I have read and agree with the venue's withdrawal policy on behalf of myself and my co-authors.